# Local Quantum Theory with Fluids in Space-Time

**Mordecai Waegell** [1,2]

1   Institute for Quantum Studies, Chapman University, Orange, CA 92866, USA; waegell@chapman.edu
2   Schmid College of Science and Technology, Chapman University, Orange, CA 92866, USA

**Abstract:** In 1948, Schwinger developed a local Lorentz-covariant formulation of relativistic quantum electrodynamics in space-time which is fundamentally inconsistent with any delocalized interpretation of quantum mechanics. An interpretation compatible with Schwinger's theory is presented, which reproduces all of the standard empirical predictions of conventional delocalized quantum theory in configuration space. This is an explicit, unambiguous, and Lorentz-covariant "local hidden variable theory" in space-time, whose existence proves definitively that such theories are possible. This does not conflict with Bell's theorem because it is a local many-worlds theory. Each physical system is characterized by a *wave-field*, which is a set of indexed piece-wise single-particle wavefunctions in space-time, each with its own coefficient, along with a memory which contains the separate local Hilbert-space quantum state at each event in space-time. Each single-particle wavefunction of a fundamental system describes the motion of a portion of a conserved fluid in space-time, with the fluid decomposing into many classical point particles, each following a world-line and recording a local memory. Local interactions between two systems take the form of local boundary conditions between the differently indexed pieces of those systems' wave-fields, with new indexes encoding each orthogonal outcome of the interaction. The general machinery is introduced, including the local mechanisms for entanglement and interference. The experience of collapse, Born rule probability, and environmental decoherence are discussed, and a number of illustrative examples are given.

**Keywords:** many worlds; local quantum theory; local hidden variable theory; relativistic quantum theory; Bell's theorem

## 1. Introduction

Despite insubstantial but influential claims from the early days of quantum theory, Bohm proved in 1952 [1] that it is possible to give a straightforward realist interpretation of quantum mechanics with particles in space-time. However, in that theory the underlying physics occurs in a higher-dimensional configuration space, resulting in explicitly nonlocal dynamics in space-time. In this article we lay out the general framework for a local realist collapse-free theory of quantum mechanics, and work through the simplest examples, with all dynamics occurring explicitly in space-time. This realizes an unachieved goal of Einstein, Schrödinger, and Lorentz, who were never satisfied with the configuration space treatment, precisely because it introduced fundamental nonlocality [2]. The new model makes identical empirical predictions to standard quantum theory, and can serve as a full replacement. This model is consistent with the Lorentz covariant Heisenberg-Schrödinger model proposed by Schwinger in 1948 [3], and restores the equivalence between the local Heisenberg and Schrödinger pictures. However, we now know from Bell's theorem [4–8] that if we wish to maintain independence of measurement settings, then this is unavoidably a theory of many local worlds [9].

It is important to emphasize here the breadth of Schwinger's accomplishment. In deriving quantum electrodynamics (QED) in parallel to Feynamn, he obtained a new Lorentz covariant state vector, defined on a single space-like hypersurface (which can be the 'present' hypersurface in at most one specific frame), with information at each point in the surface restricted to that point's past light cone (the events on a space-like hypersurface

are space-like separated in all frames, and have the same past events in their respective past light cones). He also obtained the localized Schrödinger-like dynamics that shows how this state evolves locally to the next parallel space-like hypersurface, and obtained a space-time invariant local interaction unitary for QED. This treatment is at the heart of modern particle physics, but these state vectors are completely inconsistent with the configuration-space wavefunctions in prevalent use throughout modern quantum foundations and information theory. To be very clear, the Lorentz covariant state vectors of the most precisely verified theory of modern physics are defined on space-like hypersurfaces in space-time, where each event on such a surface contains separate physical information, which can only pertain to past events within its past light cone. *The apparent nonlocality of conventional quantum theory is a mathematical artifact of projecting all of the space-like separated information from an entire Schwinger state, defined on a given 'present' hypersurface, into a single nonseparable delocalized state.* Furthermore, there is generally not enough information in such a delocalized quantum state to reconstruct the corresponding Schwinger state. Thus, the entangled wavefunctions of Copenhagen and spontaneous collapse theories, or the universal wavefunctions of Wheeler-DeWitt, Everett, Bohm, and others, are all delocalized approximations of a fundamentally local state in QED. This fact is not at all obvious, because one needs a proper local many worlds interpretation with explicit local hidden variables in space-time to make sense of empirical observations.

The present model is an attempt to interpret the empirical data from table-top quantum experiments, rather than high energy particle collisions, using the QED structure, by reconstructing the information content of Schwinger's space-time state vector using more familiar single-particle spatial wavefunctions. This turns out to be the natural theoretical framework for refining the local Schrödinger picture of the Parallel Lives interpretation of quantum mechanics [10,11], and should also be consistent with (but not identical to) the local Heisenberg picture frameworks that have been developed elsewhere [12–22]. I recommend perusing the detailed examples in the Supplemental Information as you read this article to help develop a clear idea of the local hidden variable treatment for Wigner's friend, Mach-Zehnder Interferometers, Wheeler's Delayed Choice, the Delayed Choice Quantum Eraser, and Quantum Teleportation. This model is a major paradigm shift from standard quantum theory, and while many familiar mathematical objects are still present, they have been put together in an entirely different way. The detailed examples should help to develop intuition for the new paradigm, and to put aside intuition from standard quantum theory which does not apply here.

In the present model, all (quantum) systems are comprised by pseudo-classical fluids in a single objective locally-Minkowski space-time and the classical particles in these fluids follow world-lines through that space-time. To give an explicit example, the probability density $|\psi(\vec{x})|^2$ for a standard single-electron wavefunction is re-interpreted as the local density of a literal fluid in space-time, and the conservation of probability current is re-interpreted as conservation of fluid current. Thus, a single electron with a spatially distributed wavefunction is interpreted as an entire fluid made of a countably infinite number of fluid particles, each of which is like a classical point particle on a world-line. There are many worlds *only* in the sense that there are many world-lines for the many such particles in space-time, and each particle experiences a unique perspective from its location in space-time. According to relativity theory, all empirical experiences necessarily follow from these unique local perspectives, and are fully restricted to an observer's past light cone. That is, for each observer, the 'world' is the image of the surface of that observer's past light cone. This definition makes the set of events in a 'world' Lorentz invariant. There are no global 'worlds' in this theory - there is only the one global space-time, containing many particles on world-lines, each with its own past light cone and 'world', as shown in Figure 1. To be very explicit, even though their resolutions to the measurement problem are similar, the local space-time model presented here is fundamentally different from the many-worlds theory of Everett [23,24], which is delocalized in configuration space, and describes global worlds in a particular Lorentz frame. There is no space-like hypersurface

that is observed by even one, let alone many observers, and it is a mistake to define global worlds on these hypersurfaces. This is corroborated by the fact that one cannot Lorentz transform the delocalized wavefunctions defined for these surfaces between different inertial frames (a Lorentz transform is a mapping that can act only on a 4-vector or field tensor at a single event, and the descriptions of these 4-vectors and fields are inherently separable event-by-event). All empirical data pertains to events within the observer's past light cone, and that data defines the world for that observer. Importantly, no observer's world contains the results of two space-like separated measurement events until signals have arrived to the observer from both of those events (i.e., once the observer has seen the empirical data).

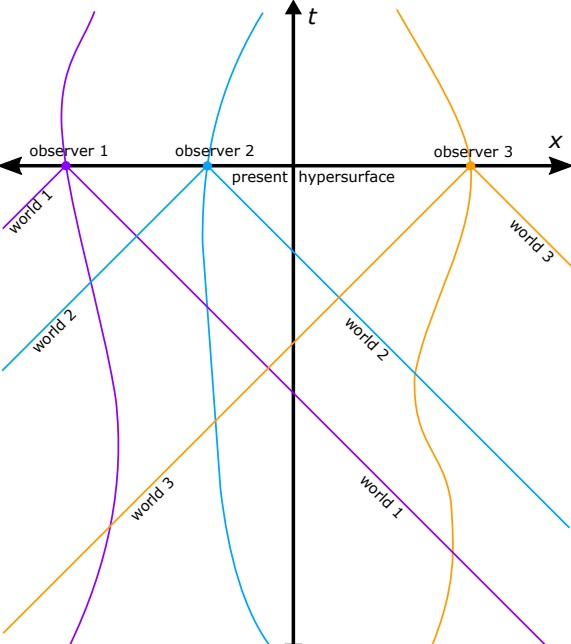

**Figure 1.** The distinct perspectives of observers on different world-lines in special relativity already provide a clear notion of many worlds. Each fluid particle has its own world in exactly this sense, and the worlds of two particles can only coincide if and when they are both present at the same event. No one observes a space-like hypersurface, so it is a mistake to define global worlds on those surfaces.

It should be noted that there are some similarities between the present model and the work of Madelung [25], and also various works on many-interacting-worlds [26–31] for a single quantum particles, but the details of the present model are distinct.

We will not be working with the individual trajectories of the classical particles in the fluids here, since we do not yet know how to choose a unique solution. The decomposition of the conserved single-particle quantum probability current $\vec{j}(\vec{x})$ into fluid streamlines where the velocity field is given by $\vec{v}(\vec{x}) = \vec{j}(\vec{x})/|\psi(\vec{x})|^2$ serves as the simplest proof-of-concept example of a viable set of trajectories - but this is not the only possible set of trajectories consistent with $\vec{j}(\vec{x})$ and $|\psi(\vec{x})|^2$.

Here we interpret the single-particle Schrödinger/Dirac equations to be continuum fluid equations consistent with coarse-graining over the trajectories of the individual particles comprising the fluid - which is also to say taking the smoothed local average over their (unknown) velocity distribution. The behavior for multiple quantum particles is completely different than in the standard treatment, which is the main focus of this article. The empirical experience of collapse and many of its consequences are explained later, but for now, the right intuition is that each fundamental quantum system comprises a conserved fluid in space-time - and it helps to keep in mind that the fluid is composed of classical particles on world-lines.

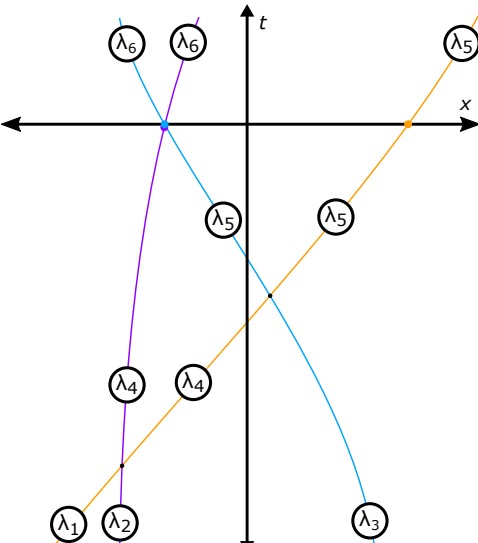

**Figure 2.** A local ballistic model is an unambiguous local hidden variable theory wherein all causal information is carried along world-lines in point-like packets. When two or more such packets meet at an event in space-time, their information undergoes a joint evolution resulting in new information that all of the packets may carry away (e.g., $\lambda_4$ results from the joint evolution of $\lambda_1$ and $\lambda_2$, etc.).

This is a *local ballistic* model of the universe, meaning all interactions are local scattering events between ballistic classical particles, and there are no nonlocal or long-range interactions or objects of any kind (i.e., all long-range effects are mediated by force-carrying particles on world-lines which undergo local collisions). In the most general local ballistic model, classical particles can carry an internal memory containing an arbitrary amount of information, and when two particles interact locally at an event in space-time, their two memories undergo a joint evolution, as shown in Figure 2. In the coarse-grained fluid picture, the set of scattering rules for such local collisions should ultimately come from the Standard Model Lagrangian, and these take the form of boundary conditions between different packets of fluid, while the memories become local properties of the continuum fluid packets. It should be emphasized that the information in the internal memories cannot be directly measured, which makes it fundamentally different from various types of practical physical memory, so unlike measurable physical memories, there is no energy density associated with these internal memories - just as there is no energy density associated with the information content of the wavefunction in standard quantum theory.

A single quantum system may comprise a superposition of many different indexed single-particle wavefunctions, each evolving independently of the others in space-time, in the absence of an interaction with another system. We can think of the indexes that delineate the different wavefunctions of a given system as belonging to its local memory, along with a separable local copy of the entangled state from which the indexes are drawn (in this model, an entangled state is just a piece of information that lives at a single event in space-time, not a delocalized nonseparable object spread across multiple events). For each system, it is the local scattering interactions with other fluid particles of the same system, with the same indexes, that produces the collective Schrödinger/Dirac wave evolution in the fluid, and interactions with other systems can result in local entangled states with more distinct indexes, and thus more distinct wavefunctions for each system.

We call the collective description of all indexed packets of a quantum system in space-time a *wave-field*. As we will show later, the wave-field for a single fundamental system is expressed as a piece-wise multi-valued wavefunction in space-time, where each indexed value evolves independently according to the single-particle Schrödinger/Dirac equation. The pieces are separated in space-time by interaction-based boundary conditions, which are the locations where the fluid particles scatter and their internal memory states synchronize

and evolve. The synchronization of the internal memory states generally increases the respective numbers of orthogonal terms, and thus also increases the number of indexed wavefunctions on the other side of the boundary. The wave-field of a system is a separable mathematical description for that system alone - even if it is entangled with other systems. The set of all wave-fields on a given space-like hypersurface should be consistent with the covariant state introduced by Schwinger.

In the non-relativistic limit, we can use Bohm's eikonal form of a set of indexed single-particle wavefunction $\psi_i = R_i e^{iS_i/\hbar}$ to elucidate the fluid picture, where $R_i^2$ is the fluid density, $S_i$ is Hamilton's principal function, and $\vec{\nabla} S_i/m$ is the local average velocity field of the particles in the fluid. Then $R_i$ and $S_i$ evolve according to the coupled continuity equation and Hamilton-Jacobi equation, which motivates the fluid picture much more clearly than the Schr/"odinger evolution. For a superposition state with multiple orthogonal terms indexed by $i$, the coefficient $a_i = |a_i|e^{i\phi_{a_i}}$ of each terms give the total proportion $|a_i|^2$ and global phase $\phi_{a_i}$ of the corresponding packet of fluid. These are the phases and proportions of the total fluid in the total wave-field, which are relevant for interference. As we will see, it is still essential that each particle in the fluid carries its own copy of the entire local state in its memory, in order to properly define the local transfer matrices at interactions. The relativistic treatment is conceptually identical, with the fluid particles moving along world-lines.

Macroscopic systems are truly composed of many fundamental single-particle systems, each with its own fluid and set of single-particle wavefunctions, but in many cases the correct intuition can be obtained by approximating the macroscopic system as a single fluid, whose particles are different copies of the whole system. This allows us to neglect the fine details of the internal local interactions between the fundamental constituents of that system.

For macroscopic systems, there is a clear preferred basis, and the indexes correspond to the experience of one outcome or another during an interaction (measurement), which ultimately gives rise to the Born rule. A system does not directly experience its own internal memory - only its index (external memory). In practice, these macroscopic external memory records are permanent (although it is possible in theory to project those systems into noncommuting bases, which would overwrite that memory).

Finally, to get some physical intuition for this model, it helps to think of each indexed wave packet as an isolated drop of fluid floating through space. This is a very nonclassical fluid, which behaves more like a gas than a liquid, allowing significant compression and rarefaction as it moves. This facilitates longitudinal waves passing through the drop, which produces familiar wave behavior. Unlike a classical gas, these waves can create zeros in the fluid density (the nodes of a stationary state, for example) - so the local scattering rules for the particles in the fluid must also be quite nonclassical. Despite this, fluid particles never cross these zeros, and the motion of the entire fluid can always be decomposed into their world-lines.

Quantum tunneling through a finite barrier highlights the nonclassicality of the fluid. As a pulse is incident upon a barrier, the interference with the reflected wave may cause temporary zeros to form in front of the barrier, and the fluid to form a series of compressed and rarefied regions, which quickly vanish as the reflected pulse moves away. Part of the packet also penetrates inside the barrier, and the probability current there is nonzero, so the fluid particles' world-lines are literally passing through the barrier and continuing on the other side - and clearly with a nonzero tunneling time.

As we will see, the complexity of the picture grows with the dimension of the usual Hilbert space, which only obscures some of the relevant features, and this is why we focus most of our attention on examples with 2-level systems. This article begins with analysis of states of one, two, and three spins in space-time, including Von Neumann measurement and Born's rule. We then give a full demonstration of the local treatment of an experimental test of Bell's theorem, treating Alice and Bob as spins. Spatial entanglement and the Stern–Gerlach experiment are also discussed. We conclude with some discussion of the

historical context of this model as well as its potential future applications. The Supplemental Information contains discussion of a number of other important experiments.

## 2. Internal and External Memory

The internal memory of a particle in the fluid of a given system has a simple general form, using standard quantum language, but it must be emphasized that this is a fully separated piece of information contained at a single point in space-time, and carried with the particle on its world-line.

First, the memory contains a product state of the initial state for its own system, and the initial states for the first interaction with any other system in its past-interaction-cone (even if these systems did not directly interact with the present one). Second, it contains a list of pair-wise unitary coupling operations from interactions between two systems, along with a list of local single-particle (kinetic) unitary evolution operators, all in temporal order for causally connected operations (in the special relativity sense). Together, these pieces of information give a standard Hilbert-space quantum state,

$$\left( \prod_i U_i \right) \bigotimes_j |\phi_j\rangle^j, \tag{1}$$

where the index $i$ includes all single-particle unitaries and pair-wise interaction unitaries $U_i$ in the past-interaction cone, and the index $j$ includes the initial state of every system $j$ in the past-interaction cone (superscripts label different systems here and throughout this paper). This form does not necessarily imply that the initial state of the universe was a product state, since it could have begun with entangled local memories of this same form.

The set of indexed single-particle wavefunctions in any product basis can be trivially extracted from this state, which is also to say that the external memory can be trivially extracted from the internal memory. Each term in the superposed state represents a separate spatial wavefunction of the system, each with its own complex coefficient, and the term itself becomes the index of that wavefunction (this treatment differs from previous discussions [11] where the external-memory-history of each particle was tracked. This turns out to needlessly complicate the formalism, so we have removed it from the present treatment).

Whenever two systems interact at an event, their past-interaction cones become identical by definition, and thus their internal memories synchronize, merging their two prior lists into a new shared set, $\{U_i\}$ and $\{|\phi_j\rangle^j\}$. For systems that are entangled within their respective (local) internal memories due to past interactions, this synchronization causes entanglement correlations to be obeyed if/when those systems interact in the future. That is essentially the complete local hidden variable theory, but many of the finer details of how this reproduces the empirical predictions of standard quantum theory are not obvious. This is a many-local-worlds theory, where each different external memory of a given macroscopic system is a different outcome experienced by the fluid particles of that system, all in one objective space-time. The worlds of two fluid particles coincide if and only if they are at the same event in space-time and they share the same external memory.

It must be emphasized that this construction can be transparently applied to any experimental analysis done using standard quantum theory and local unitary operations, and it produces the corresponding local hidden variable model of that experiment. This applies to experimental tests of Bell's theorem, delayed-choice quantum erasure, weak measurement, quantum teleportation, Wigner's friend experiments, tests of indefinite causal order, among many others for which a local hidden variable model is not obvious. We work through the essential examples here and in the Supplemental Information.

When a system undergoes a local (measurement) interaction with another system, its fluid is divided up into proportions given by the Born rule using the reduced density matrix of the synchronized internal memory state, and the external memories of each sub-part of the fluid are the different outcomes. The other interacting system has the same synchronized entangled state in its internal memory, and its fluid is thus divided up into matching proportions, with consistent external memories.

The fluid particles with different external memories have experienced different outcomes (with Born rules proportions, which also results in Born rule ensemble statistics), even though they are all still in the same space-time. Again, this constitutes many worlds only in this sense of fluid particles on many world-lines each with its own external memory. It is also empirically evident that the different fluid particles of the same systems never see or experience each other in this way (they are 'hidden' from one another), and are thus oblivious to this division of their fluid.

Macroscopic external memories of measurement outcomes are never erased in practice, so if the two macroscopic systems meet (the identity interaction) whose memories are already entangled, then the proportions of the fluids with each possible pairing of external memories in the macroscopic preferred basis are given by the Born rule for that synchronized entangled state. When this matching occurs, each product-state term in the internal memory state corresponds to a different empirical outcome for the meeting of two macroscopic systems.

Entanglement correlations are never realized at space-like separation. Instead they are realized if and when the internal memories are locally synchronized and fluids with different external memories are matched up. Prior to this matching, the overall distribution in space-time is given by the tensor product of the different reduced density matrices, and is thus completely uncorrelated.

## 3. Macroscopic Preferred Bases and Relative Collapse

A macroscopic preferred basis typically means that the environment is entangled with the system, and encodes a memory record of different outcomes that is never erased in practice. Orthogonal degrees of freedom in the terms of the internal memory state prevent interference between those terms during a local unitary interaction, and thus macroscopic systems never undergo interference effects wherein fluid particles with different external memories are mixed together as their external memories are erased and rewritten. For a macroscopic system, the fluid of copies with a given external memory may be matched and subdivided, but the macroscopic external memory record of a given copy never changes. When we think of a quantum system as having 'collapsed', this really only means that the system is entangled with the environment in some macroscopic preferred basis.

In contrast, a microscopic system can be defined as one which is not entangled with the environment in this way, which means that there is no preferred basis for observers and all of the relevant degrees of freedom can be manipulated during an experiment, so that any pair of terms can be made to interfere, and we would think of this state as remaining 'uncollapsed' during these manipulations. This means that microscopic external memories are routinely erased and rewritten, allowing interference effects. Because of this erasure/rewriting process, there is no restriction on the matching between a given copy's prior external memory, and its new external memory, provided that the final proportions match the final internal state. Previous versions of this model [11] attempted to track the external memory history of each copy, and to develop rules for the proportion of each former memory that is reset to each new memory during the interaction. While such histories must exist in this model, there can never be any macroscopic empirical evidence of them, nor of the erasure/rewriting process, so these rules cannot be uniquely determined. In the end, there is really no reason to discuss the external memories of microscopic systems in one basis or another, since they are empirically inaccessible by definition.

To make this more explicit, consider a case where a detector for a 2-level quantum system has clicked, and thus the outcomes $|0\rangle^t$ and $|1\rangle^t$ for the 2-level system has been amplified to the macroscopic scale as detector outcome states $|0\rangle^d$ and $|1\rangle^d$ and propagated through the environment such that the internal memory of an arbitrary system in the environment, including an observer, now contains a superposition of these terms in its internal memory state. The amplification has implicitly created a macroscopic preferred basis, since we never observe a superposition of two different detector clicks at the macroscopic level. To erase/rewrite this memory would require manipulating all records of the detector clicks

throughout the entire macroscopic environment, which never happens in practice. This means that the internal memory of every system in the environment is entangled with the 2-level system, with this preferred basis, and thus for those copies of a macroscopic system with $|0\rangle^d$ in their external memory, the system has 'collapsed' to state $|0\rangle^t$, since they will never meet the fluid particles of that system which had external memory with $|1\rangle^t$. Likewise, the 2-level system has 'collapsed' to $|1\rangle^t$ relative to copies of macroscopic systems with $|1\rangle^d$ in their external memory.

On the other hand, if we let two quantum systems with states $|\psi\rangle^1$ and $|\phi\rangle^2$ interact via unitary $U$ while remaining isolated from the environment (as in a quantum computer), then there are no orthogonal macroscopic states in internal memory to prevent interference between the different orthogonal terms, and thus this state is 'uncollapsed' relative to copies of a macroscopic observer. A subsequent interaction unitary $V$ is free to erase and rewrite the external memories of fluid particles of both systems *exactly because* there is no macroscopic empirical record of these external memories. For example, if $U$ is followed by $U^{-1}$, then the original product state $|\psi\rangle^1|\phi\rangle^2$ is restored.

## 4. Spins

In this model, the wave-field of an isolated Pauli spinor comprises a superposition of two indexed single-particle wavefunctions in space-time, each of which is represented by a fluid density $R_i^2(\mathbf{x}, t)$ and a principal function $S_i(\mathbf{x}, t)$ with velocity field $\vec{\nabla} S_i(\mathbf{x}, t)/m$. If two spins are entangled, then each spin comprises up to four wavefunctions in space-time. For three entangled spins, each comprises up to eight wavefunctions in space-time – one for each orthogonal term in the local internal memory state. We can work in any product basis without loss of generality, so we use the binary basis for all Pauli spinors, meaning $i = 0, 1$, corresponding to spin states $|0\rangle$ and $|1\rangle$.

For a single spin (system 1, denoted by the superscript), the two wavefunctions $a\psi_0^1(x)$ and $b\psi_1^1(x)$ correspond to the spin states $|0\rangle^1$ and $|1\rangle^1$, respectively, and it is the sum of these two probability densities that is normalized in space ($|a|^2 + |b|^2 = 1$, $\int_{-\infty}^{\infty} |\psi_i^1(x)|^2 dx = 1$). The point is, if $\psi_0^1(x) = \psi_1^1(x) = \psi^1(x)$, then the spin-position Hilbert space product state $(a|0\rangle^1 + b|1\rangle^1)\psi^1(x_1)$ in standard quantum theory is replaced in the new theory by the pair of fluid packets in 3-space $\{a\psi_0^1(x), b\psi_1^1(x)\}$. Note that if the spin and position are entangled, the description is more complicated.

We can change the spin basis used for the representation, which results in new coefficients, and a new division into two different fluids. Regardless of the basis, the different fluids undergo independent local evolution, from which it is clear that the basis is not physically relevant for the evolution.

The wave packets themselves are constructed in the local Fock basis, but we use the shorthand notation $\psi(x) \equiv \int \psi(x) a_x^\dagger |0\rangle dx$ throughout this text.

### 4.1. Two Spins

If two spins (initially in state $|0\rangle$) have interacted via a unitary $U^{12}$ (a pure spin-spin coupling, that does not entangle the spatial degrees of freedom), then to find the spatial wavefunctions of each system, we construct the 2-spin Hilbert space state $U^{12}|0\rangle^1|0\rangle^2 = \sum_{i,j=0}^1 a_{ij}|i\rangle^1|j\rangle^2$, and then treat the spin states as indexes that delineate separate spatial wavefunctions of the present system. For example, the four local states of system 1 are,

$$a_{00}|0\rangle^1_{|0\rangle^2}, \quad a_{01}|0\rangle^1_{|1\rangle^2}, \quad a_{10}|1\rangle^1_{|0\rangle^2}, \quad a_{11}|1\rangle^1_{|1\rangle^2}, \tag{2}$$

where the subscript contains an external memory state for other systems, and the four corresponding wavefunctions are

$$\begin{aligned} a_{00}\psi(x)^1_{0,|0\rangle^2}, &\quad a_{01}\psi(x)^1_{0,|1\rangle^2}, \\ a_{10}\psi(x)^1_{1,|0\rangle^2}, &\quad a_{11}\psi(x)^1_{1,|1\rangle^2}, \end{aligned} \tag{3}$$

where the first subscript is the spin external memory of the present system, and it is the sum of these four probability densities that is normalized in space ($|a_{00}|^2 + |a_{01}|^2 + |a_{10}|^2 + |a_{11}|^2 = 1$). Likewise for system 2 the local states are,

$$a_{00}|0\rangle^2_{|0\rangle^1}, \quad a_{01}|1\rangle^2_{|0\rangle^1}, \quad a_{10}|0\rangle^2_{|1\rangle^1}, \quad a_{11}|1\rangle^2_{|1\rangle^1}, \tag{4}$$

and the four corresponding wavefunctions are

$$a_{00}\psi(x)^2_{0,|0\rangle^1}, \quad a_{01}\psi(x)^2_{1,|0\rangle^1},$$
$$a_{10}\psi(x)^2_{0,|1\rangle^1}, \quad a_{11}\psi(x)^2_{1,|1\rangle^1}. \tag{5}$$

In the absence of an interaction, each of these spatial wavefunctions evolves independently according to the single-particle Schrödinger/Dirac equation. Local interactions take the form of spatial boundary conditions that connect the different spatial wavefunctions.

After the local interaction, each of them carries a copy of the information $U^{12}|0\rangle^1|0\rangle^2$ in its local memory. This process of collecting local interaction unitaries and initial states along a single world-line is the essence of the local Heisenberg treatment used in relativistic quantum field theory. It is important to emphasize that each of these copies is separate and independent from the others, and each copy encodes local information at a single point in space (a single fluid particle). Whenever two systems locally interact, the memories of the two systems first synchronize before the new interaction unitary is applied, so that they now share all unitary operations and initial states from both of their past local interaction cones. Then the new interaction unitary is added to both of them, resulting in an equal number of indexed spatial wavefunctions for each system, with matching coefficients.

Taking the simple case that the spin and spatial wavefunctions are not entangled, we have $\psi(x)^1_{0,|0\rangle^2} = \psi(x)^1_{0,|1\rangle^2} = \psi(x)^1_{1,|0\rangle^2} = \psi(x)^1_{1,|1\rangle^2} = \psi(x)^1$ and $\psi(x)^2_{0,|0\rangle^1} = \psi(x)^2_{1,|0\rangle^1} = \psi(x)^2_{0,|1\rangle^1} = \psi(x)^2_{1,|1\rangle^1} = \psi(x)^2$, and the standard quantum state of two entangled spins with separable position states in Hilbert/configuration space,

$$\psi(x_1)^1 \otimes \psi(x_2)^2 \otimes \sum_{i,j=0}^{1} a_{ij}|i\rangle^1|j\rangle^2, \tag{6}$$

is replaced in the local theory by the above set of eight fluid packets in 3-space, and the many local copies of the Hilbert-space state $U^{12}|0\rangle^1|0\rangle^2$ they carry in memory.

Note that the local memories of each spin carry all of the same information about this interaction as the entangled Hilbert space state of conventional quantum theory, and at the fine-grained scale, every fluid particle of each system carries this information on its world-line as well. The main point here is that these eight fluid packets evolving in space-time contain all of the information needed to produce the correct empirical probabilities and entanglement correlations for these systems. Because all interactions are local, we can completely replace delocalized wavefunctions in higher-dimensional spaces with the wave-field in space-time, and obtain all of the original empirical predictions. This is our local hidden variable theory.

By way of notation in the present formalism, we will use superscripts to indicate which system a spatial wavefunction belongs to, rather than subscripts on the coordinates in a single configuration space wavefunction for all systems. As shown above, all internal degrees of freedom (like spin) now correspond to additional indexed spatial wavefunctions of a given system, and entanglement with other systems results in additional spatial wavefunctions for both systems.

### 4.2. Three or More Spins

Now, suppose that system 1 interacts locally with system 3, while system 2 is not involved, and does not change in any way. The interaction unitary is $V^{13}$ and the initial state of system 3 is $|0\rangle^3$, and system 3 carries no other relevant memory. First, the two systems

synchronize memory to $U^{12}|0\rangle^1|0\rangle^2|0\rangle^3$, so system 3 splits into four indexed wavefunction, whose coefficients match those of systems 1. Then $V^{13}$ is added to both memories, resulting in state,

$$V^{13}U^{12}|0\rangle^1|0\rangle^2|0\rangle^3 = \sum_{i,j,k,l=0}^{1} a_{ij}b_{ikl}|k\rangle^1|l\rangle^3|j\rangle^2$$

$$= \sum_{j,k,l=0}^{1} c_{jkl}|k\rangle^1|l\rangle^3|j\rangle^2. \tag{7}$$

From this, we see that there are eight local spin states for each system,

$$\left\{ c_{jkl}|k\rangle^1_{|l\rangle^3|j\rangle^2} \right\}, \quad \left\{ c_{jkl}|l\rangle^3_{|k\rangle^1|j\rangle^2} \right\}, \tag{8}$$

where in either case the other two systems are treated as indexes, and thus eight spatial wavefunctions for each system as well,

$$\left\{ c_{jkl}\psi(x)^1_{k,|l\rangle^3|j\rangle^2} \right\}, \quad \left\{ c_{jkl}\psi(x)^3_{l,|k\rangle^1|j\rangle^2} \right\}. \tag{9}$$

Now, systems 2 and 3 have not interacted, but there are now entangled within the local memory of system 3, which will effect what happens if they interact in the future. Let us consider that case next.

Systems 2 and 3 now interact via unitary $W^{23}$. First, their memories are synchronized to $V^{13}U^{12}|0\rangle^1|0\rangle^2|0\rangle^3$, which splits the four indexed wavefunctions of system 2 into eight, such that the entanglement correlations between systems 2 and 3 become physically manifest. Note that if two interacting systems already share some unitaries or initial states in memory, they will necessarily match, and so the two memories can be simply be merged, as in this case (before the interaction, system 2 had $U^{12}|0\rangle^1|0\rangle^2$).

Now the new interaction unitary is added to both memories, resulting in

$$W^{23}V^{13}U^{12}|0\rangle^1|0\rangle^2|0\rangle^3 = \sum_{j,k,l=0}^{1} d_{jkl}|k\rangle^1|l\rangle^3|j\rangle^2, \tag{10}$$

and eight new wavefunctions for systems 2 and 3. The eight wavefunctions of system 1 are not involved in this interaction, and are unchanged. This series of interactions are shown in Figure 3, along with the internal memory being carried by each system. Hopefully the general picture for larger numbers of spins is clear at this point. The Supplemental Information contains the detailed treatment of several important experiments.

Note that after any of the interactions in Figure 3, the proportion of fluid with each orthogonal external memory in any basis is given by the Born rule, and exactly matches standard quantum theory for the same initial states and unitary interactions. Over an ensemble of experimental trials, empirical probabilities emerge which match the proportions of fluid with each orthogonal external memory in the macroscopic preferred basis, and this exactly reproduces the probabilities predicted by standard delocalized quantum theory.

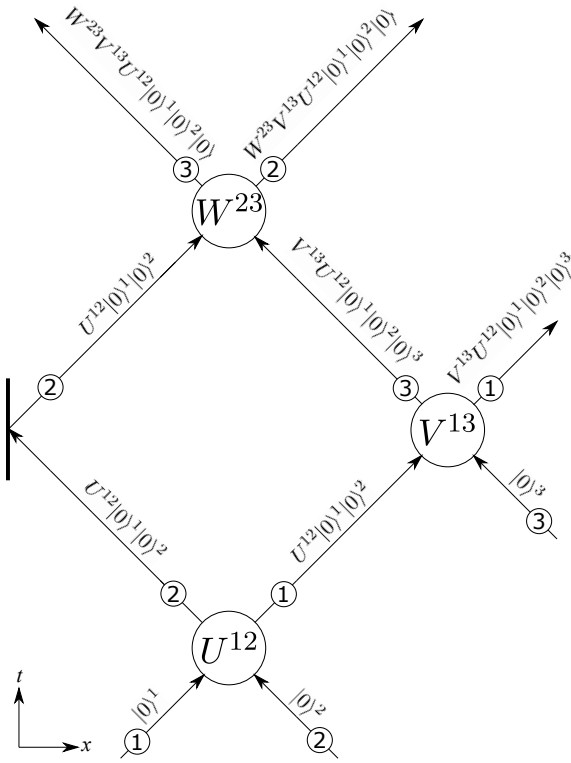

**Figure 3.** An illustration of the local hidden variables of the present model, showing three systems undergoing a series of local spin-spin interactions in space-time. The memories of systems 1 and 2 synchronize when they meet, and the interaction unitary $U^{12}$ is added to both. Then the memories of systems 1 and 3 synchronize when they meet, and the interaction unitary $V^{13}$ is added to both. The memory of system 2 is unaffected by this space-like separated interaction. Finally, the memories of systems 2 and 3 are synchronized when they meet, the interaction unitary $W^{23}$ is added to both, and the expected entanglement correlations between those systems are obeyed. The internal states can be expanded in any product basis to give the local set of single-particle wavefunctions in space-time indexed by the external memories in that basis, where the interaction unitaries define boundary conditions connecting the pre-interaction fluids to the post-interaction fluids. If there is a macroscopic preferred basis, then expanding the internal memory in that basis gives the set of different external memories experienced by different copies of macroscopic observers in space-time. If one considers a Schwinger state on any space-like hypersurface that cuts across this diagram, it is easy to see what information is encoded at each event on that surface, and to verify that this information only pertains to that event's past light cone. It is also easy to see that the Schwinger state on the 'present' surface contains too much information to be reconstructed from a standard delocalized quantum state.

### *4.3. Local Entanglement*

What remains is to show how two systems interact locally and become entangled in this way. We will begin with the simplest possible example, which is also quite illustrative. We expect the general theory to contain only one type of coupling potential, and this is of the form $\delta(\vec{x}_1 - \vec{x}_2)V$, where $V$ is a general space-time-independent potential. This says that when two systems meet at an event in space-time, the potential $V$ produces the local scattering between them, via the unitary $U = e^{-iV/\hbar}$. The new states are written into the memories of the fluid particles as this happens, causing them to separate into more distinct fluids than before. This general potential should be uniform throughout space-time, and encompass all possible scattering events between all types of quantum systems. In other words, all Standard Model particle interactions should be encoded in $V$. The specific formalism for deriving a local unitary operation that acts only at a single event is highly nontrivial, and subject to ongoing research.

Because this model does not support long-range interactions, it is relatively complicated to recover Coulomb-potential based interactions between charges, which are mediated by massless force carriers. To demonstrate the general mechanism, we have restricted ourselves to a *gedanken* experiment with just two quantum systems in space-time, where the only coupling potential is a spin–spin interaction – thus $U = e^{-iV/\hbar}$ is some $4 \times 4$ 2-spin matrix. With this potential, the spatial wavefunctions never change or become entangled with the spins. Thus, if the two systems are incident upon one another, the fluid packets pass through the interaction without being deflected or deformed, but the spin states interact locally as this occurs, causing the fluid to acquire new indexes that separate it into more distinct packets than before - each moving independently but identically. These local interactions are where the internal memory states become entangled, so it is still appropriate to say that two systems became entangled during this interaction, even though this model has no delocalized entangled state in Hilbert/configuration space.

We now consider such an entangling interaction for two spins that begin in a separable state, each with two wavefunctions in space-time, $a_1\psi_0^1(x,t)$ and $b_1\psi_1^1(x,t)$, and $a_2\psi_0^2(x,t)$ and $b_2\psi_1^2(x,t)$, respectively. They can only interact locally, and thus the only reason they have not interacted is that they have no overlapping support. In fact, there must be a boundary point $x_{12}$ that separates their supports. For this simple one-dimensional example, we will begin with spin 1 located fully to the left of $x_{12}$ and propagating towards spin 2, which is fully to the right of $x_{12}$. In this example, once their supports begin to overlap, the spins will directly interact via a 4-dimensional unitary $U$ (strictly speaking, it need not be unitary so long as it is norm-preserving), which maps the two pre-interaction wavefunctions of each system into its four post-interaction wavefunctions. Note that for spin 1 the two pre-interaction wavefunctions are only supported at $x \leq x_{12}$, while the four post-interaction wavefunctions are only supported at $x \geq x_{12}$ (since the wave-packets continue to propagate with the same momentum). From here, it is clear that $U$ simply defines the boundary conditions that connect these six wavefunctions at $x_{12}$ (four post-interactions wavefunctions on one side to two pre-interaction wavefunctions on the other side), and the fully normalized piece-wise wave-field $|\Psi(x,t)\rangle^s$ of each system includes contributions from all six, and all twelve for both systems. The situation is shown in Figure 4.

### 4.4. The Interaction Boundary

The next important detail we need to examine is the actual location $x_{12}$ between the systems, which is not a fixed boundary at all, but rather a dynamic one that moves in time depending on the shapes of the two incident systems' wavefunctions (in 3D this is a dynamic boundary surface). The boundary is defined by a special rule that applies to all entanglement couplings in this model - the fluid flux of the two systems across the boundary must be equal and opposite. For any two normalized wavefunctions $\psi(x)$ and $\phi(x)$, there is always a boundary point where,

$$\int_{-\infty}^{x_{12}} |\psi^1(x)|^2 dx = \int_{x_{12}}^{\infty} |\psi^2(x)|^2 dx, \tag{11}$$

and

$$\int_{-\infty}^{x_{12}} |\psi^2(x)|^2 dx = \int_{x_{12}}^{\infty} |\psi^1(x)|^2 dx. \tag{12}$$

The initial value of the boundary can be found in this way (ideally when the two packets are well-separated), and then it moves according to,

$$\dot{x}_{12}(t) = \frac{j^1(x_{12},t) + j^2(x_{12},t)}{|\psi^1(x_{12},t)|^2 + |\psi^2(x_{12},t)|^2}, \tag{13}$$

where $j^s$ is the current density of each fluid. This equal-and-opposite flux condition guarantees that an equal amount of fluid from each system is always crossing the boundary in a given time.

This condition is required to guarantee that the $a_{00}$ in $a_{00}\psi^1_{0_{0^1 0^2}}$ is the same as in $a_{00}\psi^2_{0^1 0^2}$, the $a_{01}$ in $a_{01}\psi^1_{0,|1\rangle^2}$ is the same as the $a_{01}$ in $a_{01}\psi^2_{|0\rangle^1}$, etc. As they cross the interaction boundary, the fluid particles of both systems acquire all of the 2-spin entanglement information. Importantly, these are independent copies of the coefficients, in different memory records, and local interventions on one copy have no nonlocal effect on other copies.

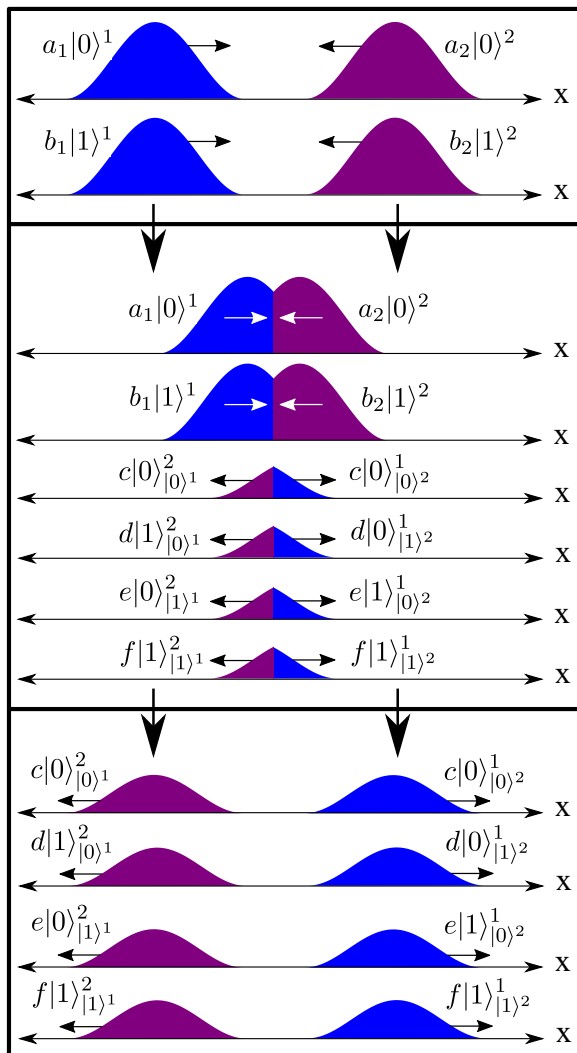

**Figure 4.** Three frames showing the local interaction process as two particles in one dimension pass through each other, with only their spins interacting. The spatial density $|\psi(x)_i|^2$ of each fluid pulse is shown, each indexed by past interactions, for the particular case that $|a_1|^2 = |b_1|^2 = |a_2|^2 = |b_2|^2 = 1/2$ and $|c|^2 = |d|^2 = |e|^2 = |f|^2 = 1/4$. The piece-wise wave-field $|\Psi(x,t)\rangle^s$ of each system formally includes all six wavefunctions as separated at the dynamic boundary $x_{12}(t)$ (stationary in this example) which all occupy the same space-time. Also consider this example in a boosted Lorentz frame, where the boundary is moving such that the fluid fluxes of the two systems are equal and opposite.

In 3D, there is no longer a unique boundary surface for two normalized functions, so an initial boundary must be assumed. The motion of this boundary is then defined locally such that as two systems move together, an equal amount of fluid from each crosses per

unit time. At the fine-grained scale, the boundary corresponds to the locations where fluid particles of the two systems are locally scattering and synchronizing memory.

A boundary like this exists between every pair of systems. It goes beyond the scope of the present article, but once the spatial degrees of freedom of a system are entangled, there can generally be different interaction boundaries for different wavefunctions of a given pair of systems.

Finally, the hard boundary presented here may be a proof of concept, rather than a physically correct rule. A possible generalization is that when two fluids meet, only some fraction of them interacts, and the remainders simply continue in their pre-interaction states. This would effectively smear the boundary between the pre- and post-interaction wavefunctions, but it would also mean that two fluids can never fully switch into their post-interaction states.

While the mathematics of dynamical boundaries is quite complicated, we expect that the exact details can be cleanly extracted from Schwinger's theory in the future. The exact details of the boundary are not important in many practical situations anyway, so for the purpose of this article, we will simply assume the existence of a hard boundary.

*4.5. Boundary Conditions*

To obtain the boundary condition at $x_{12}$, we consider the action of a general norm-preserving transformation matrix on a general product state of two spins. The actual choice of basis for this analysis is completely arbitrary. The matrix can be expanded as

$$U^{12} = \sum_{i,j,k,l \in [0,1]} u_{ijkl} |i\rangle^1 |j\rangle^2 \langle k|^1 \langle l|^2, \tag{14}$$

and then

$$U^{12}\big(a_1 a_2 |0\rangle^1 |0\rangle^2 + a_1 b_2 |0\rangle^1 |1\rangle^2 \tag{15}$$

$$+ b_1 a_2 |1\rangle^1 |0\rangle^2 + b_1 b_2 |1\rangle^1 |1\rangle^2\big) =$$

$$\big(u_{0000} a_1 a_2 + u_{0001} a_1 b_2 + u_{0010} b_1 a_2 + u_{0011} b_1 b_2\big) |0\rangle^1 |0\rangle^2$$

$$+ \big(u_{0100} a_1 a_2 + u_{0101} a_1 b_2 + u_{0110} b_1 a_2 + u_{0111} b_1 b_2\big) |0\rangle^1 |1\rangle^2$$

$$+ \big(u_{1000} a_1 a_2 + u_{1001} a_1 b_2 + u_{1010} b_1 a_2 + u_{1011} b_1 b_2\big) |1\rangle^1 |0\rangle^2$$

$$+ \big(u_{1100} a_1 a_2 + u_{1101} a_1 b_2 + u_{1110} b_1 a_2 + u_{1111} b_1 b_2\big) |1\rangle^1 |1\rangle^2$$

$$= c |0\rangle^1 |0\rangle^2 + d |0\rangle^1 |1\rangle^2 + e |1\rangle^1 |0\rangle^2 + f |1\rangle^1 |1\rangle^2$$

This allows us to define the two $4 \times 2$ transfer matrices $T_1$ and $T_2$ that map the two pre-interaction wavefunctions of each system onto its four post-interaction wavefunctions,

$$T_1^U = U^{12}\big(a_2 |0\rangle^2 + b_2 |1\rangle^2\big)$$

$$= \sum_{i,j,k \in [0,1]} (u_{ijk0} a_2 + u_{ijk1} b_2) |i\rangle^1 |j\rangle^2 \langle k|^1$$

$$= \begin{bmatrix} u_{0000} a_2 + u_{0001} b_2 & u_{0010} a_2 + u_{0011} b_2 \\ u_{0100} a_2 + u_{0101} b_2 & u_{0110} a_2 + u_{0111} b_2 \\ u_{1000} a_2 + u_{1001} b_2 & u_{1010} a_2 + u_{1011} b_2 \\ u_{1100} a_2 + u_{1101} b_2 & u_{1110} a_2 + u_{1111} b_2 \end{bmatrix},$$

and

$$T_2^U = U^{12}\big(a_1 |0\rangle^1 + b_1 |1\rangle^1\big)$$

$$= \sum_{i,j,l \in [0,1]} (u_{ij0l}a_1 + u_{ij1l}b_1)|i\rangle^1 |j\rangle^2 \langle l|^2$$

$$= \begin{bmatrix} u_{0000}a_1 + u_{0010}b_1 & u_{0001}a_1 + u_{0011}b_1 \\ u_{0100}a_1 + u_{0110}b_1 & u_{0101}a_1 + u_{0111}b_1 \\ u_{1000}a_1 + u_{1010}b_1 & u_{1001}a_1 + u_{1011}b_1 \\ u_{1100}a_1 + u_{1110}b_1 & u_{1101}a_1 + u_{1111}b_1 \end{bmatrix}.$$

Because $U^\dagger = U^{-1}$, we also have

$$\left(T_s^U\right)^\dagger T_s^U = \hat{I}_s, \tag{16}$$

where $\hat{I}_s$ is the identity for system $s$ alone. Finally, if $U$ is expanded into outer products then the $T$s can be expressed using the subscripts and without using matrices (see the Bell Test example below).

It is clear that the local state of the other spin appears in each transfer matrix, which makes perfect sense given that this is a local interaction between the two spins.

We can read off the coupled boundary conditions for the four post-interaction wavefunctions of each system as,

$$\begin{bmatrix} \tilde{\psi}_{0,|0\rangle^2}^1(x_{12}(t),t) \\ \tilde{\psi}_{0,|1\rangle^2}^1(x_{12}(t),t) \\ \tilde{\psi}_{1,|0\rangle^2}^1(x_{12}(t),t) \\ \tilde{\psi}_{1,|1\rangle^2}^1(x_{12}(t),t) \end{bmatrix} = T_1^U \begin{bmatrix} \tilde{\psi}_0^1(x_{12}(t),t) \\ \tilde{\psi}_1^1(x_{12}(t),t) \end{bmatrix}, \tag{17}$$

and

$$\begin{bmatrix} \tilde{\psi}_{0,|0\rangle^1}^2(x_{12}(t),t) \\ \tilde{\psi}_{0,|1\rangle^1}^2(x_{12}(t),t) \\ \tilde{\psi}_{1,|0\rangle^1}^2(x_{12}(t),t) \\ \tilde{\psi}_{1,|0\rangle^1}^2(x_{12}(t),t) \end{bmatrix} = T_2^U \begin{bmatrix} \tilde{\psi}_0^2(x_{12}(t),t) \\ \tilde{\psi}_1^2(x_{12}(t),t) \end{bmatrix}, \tag{18}$$

and for the pre-interaction wavefunctions as,

$$\begin{bmatrix} \tilde{\psi}_0^1(x_{12}(t),t) \\ \tilde{\psi}_1^1(x_{12}(t),t) \end{bmatrix} = (T_1^U)^\dagger \begin{bmatrix} \tilde{\psi}_{0,|0\rangle^2}^1(x_{12}(t),t) \\ \tilde{\psi}_{0,|1\rangle^2}^1(x_{12}(t),t) \\ \tilde{\psi}_{1,|0\rangle^2}^1(x_{12}(t),t) \\ \tilde{\psi}_{1,|1\rangle^2}^1(x_{12}(t),t) \end{bmatrix}, \tag{19}$$

and

$$\begin{bmatrix} \tilde{\psi}_0^2(x_{12}(t),t) \\ \tilde{\psi}_1^2(x_{12}(t),t) \end{bmatrix} = (T_2^U)^\dagger \begin{bmatrix} \tilde{\psi}_{0,|0\rangle^1}^2(x_{12}(t),t) \\ \tilde{\psi}_{0,|1\rangle^1}^2(x_{12}(t),t) \\ \tilde{\psi}_{1,|0\rangle^1}^2(x_{12}(t),t) \\ \tilde{\psi}_{1,|1\rangle^1}^2(x_{12}(t),t) \end{bmatrix}, \tag{20}$$

where the $\tilde{\psi}$ are general un-normalized individual wavefunctions for each index.

These reduce back to simple mappings between the spin coefficients, since all of the normalized packets are identical, so the transfer matrices really only produce the coefficients $c$, $d$, $e$, and $f$, and show how $a_s$ and $b_s$ define them.

The piece-wise multivalued wave-fields of each system are,

$$|\Psi(x,t)\rangle^1 = \begin{cases} \begin{array}{ll} a_1\psi_0^1(x,t) \\ b_1\psi_1^1(x,t) & x \le x_{12}(t) \\ \\ c\psi_{0,|0\rangle^2}^1(x,t) \\ d\psi_{0,|1\rangle^2}^1(x,t) \\ e\psi_{1,|0\rangle^2}^1(x,t) & x > x_{12}(t) \\ f\psi_{1,|1\rangle^2}^1(x,t) \end{array} \end{cases} \tag{21}$$

and

$$|\Psi(x,t)\rangle^2 = \begin{cases} \begin{array}{ll} a_2\psi_0^2(x,t) \\ b_2\psi_1^2(x,t) & x > x_{12}(t) \\ \\ c\psi_{0,|0\rangle^1}^2(x,t) \\ d\psi_{1,|0\rangle^1}^2(x,t) \\ e\psi_{0,|1\rangle^1}^2(x,t) & x \le x_{12}(t) \\ f\psi_{1,|1\rangle^1}^2(x,t) \end{array} \end{cases} \tag{22}$$

Since all six spatial wavefunctions of each system are identical and normalized, as are the coefficients in each region ($|a_s|^2 + |b_s|^2 = 1$ and $|c|^2 + |d|^2 + |e|^2 + |f|^2 = 1$), we can verify that the wave-field of each system describes a conserved fluid distribution in space-time.

### 4.6. Von Neumann Measurement and the Born Rule

We can simplify this example to illustrate the role of local entanglement during the measurement process and the experience of collapse with Born rule probability [32] in the new fluid picture.

For the Von Neumman measurement [33], we keep the initial state of spin 1, expressed as $(a_1|0\rangle^1 + b_1|1\rangle^1)\psi^1(x_1)$ in the conventional theory and $\{a_1\psi_0^1(x), b_1\psi_1^1(x)\}$ in the present theory, but set the initial state of spin 2 to the 'ready' state $|0\rangle^2\psi^2(x_2)$, meaning the we have only one spatial wavefunction $\psi_0^2(x)$ for spin 2 in this basis ($a_2 = 1$, $b_2 = 0$).

For a projective measurement, the unitary is then $U^{12} = \text{CNOT}$ [34], with spin 1 as the control qubit, which produces the standard entangled state,

$$(a_1|0\rangle^1|0\rangle^2 + b_1|1\rangle^1|1\rangle^2)\,\psi^1(x_1)\psi^2(x_2) \tag{23}$$

in the conventional theory. In the present theory, this means each system is carrying the local memory state $U^{12}(a_1|0\rangle^1 + b_1|1\rangle^1)|0\rangle^2$ and the corresponding set of four spatial wavefunctions

$$a_1\psi_{0,|0\rangle^2}^1(x), \quad b_1\psi_{1,|1\rangle^2}^1(x),$$

$$a_1\psi_{0,|0\rangle^1}^2(x), \quad b_1\psi_{1,|1\rangle^1}^2(x), \tag{24}$$

with a fraction $|a_1|^2$ of the particles in the spin 2 fluid recording the outcome $|0\rangle^1$ into their external memories, and fraction $|b_2|^2$ recording $|1\rangle^1$ (see Figure 5). The external memories of each particle in the fluid also define the experience of the particle, and thus from the perspective of each particle in the fluid of spin 2, spin 1 seems to collapse into one of its eigenstates or the other. Furthermore, in a large ensemble of identically prepared runs, spin 2 will experience $|0\rangle^1$ with relative probability $|a_1|^2$ and $|1\rangle^1$ with relative probability $|b_1|^2$, thus satisfying the Born rule.

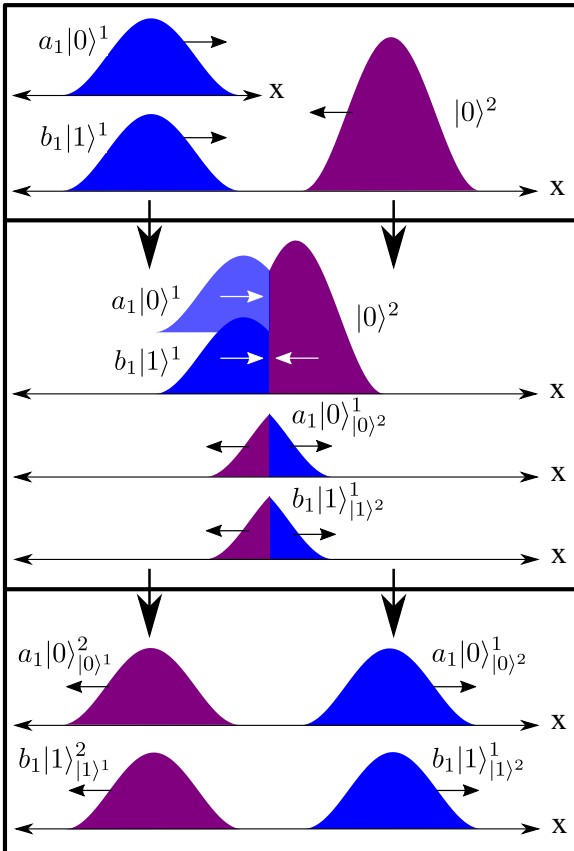

**Figure 5.** Three frames showing the local interaction process as two particles in one dimension pass through each other, with only their spins interacting. The interaction is a Von Neumann measurement of the binary basis, where system 2 is the pointer, which starts in 'ready' state $|0\rangle^2$. The spatial density $|\psi(x)_i|^2$ of each fluid pulse is shown, for the particular case that $|a_1|^2 = |b_1|^2$, each indexed by past interactions.

To round out the example, the two transfer matrices are

$$T_1^U = U^{12}|0\rangle^2 = \begin{bmatrix} 1 & 0 \\ 0 & 0 \\ 0 & 0 \\ 0 & 1 \end{bmatrix},$$

(25)

$$T_2^U = U^{12}(a_1|0\rangle^1 + b_1|1\rangle^1) = \begin{bmatrix} a_1 & 0 \\ 0 & a_1 \\ 0 & b_1 \\ b_1 & 0 \end{bmatrix},$$

and we have,

$$T_1^U(a_1|0\rangle^1 + b_1|1\rangle^1) = (a_1|0\rangle^1_{|0\rangle^2} + b_1|1\rangle^1_{|1\rangle^2}),$$

(26)

and

$$T_2^U|0\rangle^2 = (a_1|0\rangle^2_{|0\rangle^1} + b_1|1\rangle^2_{|1\rangle^1}),$$

(27)

which shows why we only have two nonzero wavefunctions for each system instead of four.

We can use these relations to define the boundary conditions at $x_{12}$ as the packets pass through one another,

$$\tilde{\psi}^1_{0,|0\rangle^2}(x_{12}(t),t) = \tilde{\psi}^1_0(x_{12}(t),t),$$

$$\tilde{\psi}^1_{1,|1\rangle^2}(x_{12}(t),t) = \tilde{\psi}^1_1(x_{12}(t),t),$$

$$\tilde{\psi}^1_{1,|1\rangle^2}(x_{12}(t),t) = \tilde{\psi}^1_{0,|0\rangle^2}(x_{12}(t),t) = 0,$$

$$\tilde{\psi}^2_{0,|0\rangle^1}(x_{12}(t),t) = a_1\tilde{\psi}^2_0(x_{12}(t),t),$$

$$\tilde{\psi}^2_{1,|1\rangle^1}(x_{12}(t),t) = b_1\tilde{\psi}^2_0(x_{12}(t),t), \tag{28}$$

$$\tilde{\psi}^2_{1,|0\rangle^1}(x_{12}(t),t) = \tilde{\psi}^2_{0,|1\rangle^1}(x_{12}(t),t) = 0,$$

$$\tilde{\psi}^2_0(x_{12}(t),t) = a_1^*\tilde{\psi}^2_{0,|0\rangle^1}(x_{12}(t),t) + b_1^*\tilde{\psi}^2_{1,|1\rangle^1}(x_{12}(t),t),$$

$$\tilde{\psi}^2_1(x_{12}(t),t) = a_1^*\tilde{\psi}^2_{0,|1\rangle^1}(x_{12}(t),t) + b_1^*\tilde{\psi}^2_{1,|0\rangle^1}(x_{12}(t),t),$$

along with the spatial derivative of these expressions evaluated at the boundary.

Each of the spatial wavefunctions $\psi(x,t)$ evolves under its own single-particle Schrödinger/Dirac equation, and all interactions occur via the boundary conditions, which automatically produce the correct splitting into more wave packets.

In summary, by treating all spin-spin interaction unitaries as local boundary conditions, and otherwise allowing each of the indexed fluids in space-time to evolve independently, we obtain the correct multiparticle quantum dynamics (without any delocalized Hilbert/configuration space evolution).

*4.7. Synchronization and General Transfer Matrices*

We have considered an interaction between two systems which were not already entangled with any other systems, and so no synchronization was necessary prior to applying the interaction unitary. However, in general, the synchronization process will extend the unitary operation being applied to each system. The transfer matrices come from the overall unitary operation, and so they are not generally $4 \times 2$. Note that the synchronization unitary is applied even if the interaction unitary is identity. Nevertheless, the construction above gives the right idea for how to construct the transfer matrices and the corresponding boundary conditions for the general case.

As an example of synchronization, let us return to the interaction between systems 1 and 3 in Section 4.1. The unitary synchronization operation $S$, when system 1 interacts interacts with system 3 results in updating the memory of system 3 from $|0\rangle^3$ to $U^{12}|0\rangle^1|0\rangle^2|0\rangle^3$, and thus the synchronization matrix is $S_3 = U^{12}I^3|0\rangle^1|0\rangle^2$, where the identity $I^3$ has been added to emphasize that this is an operation on system 3. In this case, all the operation does is introduce additional indexes that separate system 3 into four identical spatial wavefunctions. Applying $S_1 = I^{12}|0\rangle^3$ to the memory of system 1 does not change number of distinct indexes, so there are still four spatial wavefunctions.

After the synchronization, the interaction $V^{13}$ is added to the memory of both systems, resulting in the $8 \times 2$ transfer matrix $T_3^{VS} = V^{13}S_3 = V^{13}U^{12}|0\rangle^1|0\rangle^2$ for system 3. This defines the boundary condition between the two initial wavefunctions, and the eight resulting wavefunctions of system 3. For system 1, the $8 \times 4$ transfer matrix is $T_1^{VS} = V^{13}S_1 = V^{13}I^2|0\rangle^3$, which defines the boundary conditions between the four initial and eight resulting wavefunctions of system 1.

When using the single-particle unitary approximation, the transfer matrix is simply identical to the unitary, with identity matrices tacked on for other systems in the local memory.

As we can see, branching is never global in the new model. New branchings arise from the creation of new indexes during local interactions, and each branching spreads via synchronization from the systems where it originates to any other systems they interact with, and then to other systems that those interact with, and so on, in a chain that eventually applies that branching to the entire environment. This is the mechanism of decoherence in the new model, and explains the emergence of classical macroscopic experiences in dense and frequently-interacting systems.

This also explains how thought experiments like Schrödinger's cat [35], and Wigner's friend [36,37] are resolved as sequences of local interactions where branching is spread from one system to the next (see the Supplemental Information).

It is also worth noting that this mechanism produces the correct empirical experience of collapse after a measurement, because if the same two systems interact again via identity, or the measurement is repeated using a newly prepared device, the indexes all stay unchanged, meaning each observer sees the same outcome as before. If this were not the case, then the observer would in general experience violation of natural conservation laws of the dynamic quantities (energy, momentum, etc.) involved in the interaction. Even if the other system has participated in another interaction before being measured again, the results will still be consistent with the collapsed state having undergone that operation, as expected when preparing a state, and applying a unitary to it.

## 5. Bell Test

As a final example of the model, and also to demonstrate the full local treatment of entanglement, we go through a simple gedanken example of a test of Bell's theorem. This is the Mermin-Wigner test [38,39], where Alice and Bob each choose to measure their spin in one of three equally spaced directions in the $zx$-plane of the Bloch sphere.

We will begin with systems 1 and 2 having locally interacted to form the anticorrelated Bell state of two spins, $\frac{1}{\sqrt{2}}\left(|0\rangle^1|1\rangle^2 - |1\rangle^1|0\rangle^2\right)\psi^1(x_1)\psi^2(x_2)$ in the old quantum theory, and in the present theory, both systems carry the local states $U^{12}|0\rangle^1|0\rangle^2 = \frac{1}{\sqrt{2}}\left(|0\rangle^1|1\rangle^2 - |1\rangle^1|0\rangle^2\right)$ in their memory, and the four corresponding spatial wavefunctions are:

$$\frac{1}{\sqrt{2}}\psi^1_{0,|1\rangle^2}(x), \quad -\frac{1}{\sqrt{2}}\psi^1_{1,|0\rangle^2}(x),$$

$$-\frac{1}{\sqrt{2}}\psi^2_{0,|1\rangle^1}(x), \quad \frac{1}{\sqrt{2}}\psi^2_{1,|0\rangle^1}(x). \tag{29}$$

By symmetry, there are only two distinct types of measurement - those with parallel settings and those with nonparallel settings, so we only need to consider one example of each type. In both cases, Alice measures system 1 in the binary basis, while in Case 1, Bob measures system 2 in the binary basis, and in Case 2, Bob measures in the basis,

$$|\phi^+\rangle^2 = \tfrac{1}{2}\left(|0\rangle^2 + \sqrt{3}|1\rangle^2\right),$$

$$|\phi^-\rangle^2 = \tfrac{1}{2}\left(\sqrt{3}|0\rangle^2 - |1\rangle^2\right). \tag{30}$$

We treat Alice and Bob as 2-level systems in this analysis, for simplicity, and because it gets the right point across. To complete the experiment and obtain the entanglement correlations, Alice and Bob meet and interact via the identity.

*5.1. Case 1*

Alice begins in state $|0\rangle^A$ with a single spatial wavefunction $\psi_0^A(x)$ and her measurement is a $V^{1A} = $ CNOT gate with system 1 as the control and Alice as the target. The local state carried in memory synchronizes and updates to .

$$V^{1A}U^{12}|0\rangle^1|0\rangle^2|0\rangle^A = \frac{1}{\sqrt{2}}\left(|0\rangle^1|1\rangle^2|0\rangle^A - |1\rangle^1|0\rangle^2|1\rangle^A\right), \tag{31}$$

resulting in the four spatial wavefunctions,

$$\frac{1}{\sqrt{2}}\psi^1_{0,|0\rangle^A|1\rangle^2}(x), \quad -\frac{1}{\sqrt{2}}\psi^1_{1,|1\rangle^A|0\rangle^2}(x),$$

$$\frac{1}{\sqrt{2}}\psi^A_{0,|0\rangle^1|1\rangle^2}(x), \quad -\frac{1}{\sqrt{2}}\psi^A_{1,|1\rangle^1|0\rangle^2}(x). \tag{32}$$

The situation is symmetric for Bob and system 2, with local state,

$$W^{2B}U^{12}|0\rangle^1|0\rangle^2|0\rangle^B \tag{33}$$

$$= \frac{1}{\sqrt{2}}\left(|0\rangle^1|1\rangle^2|1\rangle^B - |1\rangle^1|0\rangle^2|0\rangle^B\right),$$

resulting in the four wavefunctions,

$$-\frac{1}{\sqrt{2}}\psi^2_{0,|0\rangle^B|1\rangle^1}(x), \quad \frac{1}{\sqrt{2}}\psi^2_{1,|1\rangle^B|0\rangle^1}(x),$$

$$-\frac{1}{\sqrt{2}}\psi^B_{0,|1\rangle^1|0\rangle^2}(x), \quad \frac{1}{\sqrt{2}}\psi^B_{1,|0\rangle^1|1\rangle^2}(x). \tag{34}$$

Now, Alice and Bob meet (the identity transformation), and their local memories synchronize to

$$V^{1A}W^{2B}U^{12}|0\rangle^1|0\rangle^2|0\rangle^A|0\rangle^B \tag{35}$$

$$= \frac{1}{\sqrt{2}}\left(|0\rangle^1|1\rangle^2|0\rangle^A|1\rangle^B - |1\rangle^1|0\rangle^2|1\rangle^A|0\rangle^B\right),$$

which results in the four wavefunctions,

$$\frac{1}{\sqrt{2}}\psi^A_{0,|0\rangle^1|1\rangle^2|1\rangle^B}(x), \quad -\frac{1}{\sqrt{2}}\psi^A_{1,|1\rangle^1|0\rangle^2|0\rangle^B}(x),$$

$$-\frac{1}{\sqrt{2}}\psi^B_{0,|1\rangle^1|0\rangle^2|1\rangle^A}(x), \quad \frac{1}{\sqrt{2}}\psi^B_{1,|0\rangle^1|1\rangle^2|0\rangle^A}(x). \tag{36}$$

We can now see that any fluid particle of Alice that experienced system 1 in state $|0\rangle^1$ ($|1\rangle^1$) also meets a fluid particle of Bob that experienced system 2 in state $|1\rangle^2$ ($|0\rangle^2$), and thus from the perspectives of all Alices and Bobs, the correct entanglement correlations for the Bell state have been obeyed. The steps are shown in Figure 6.

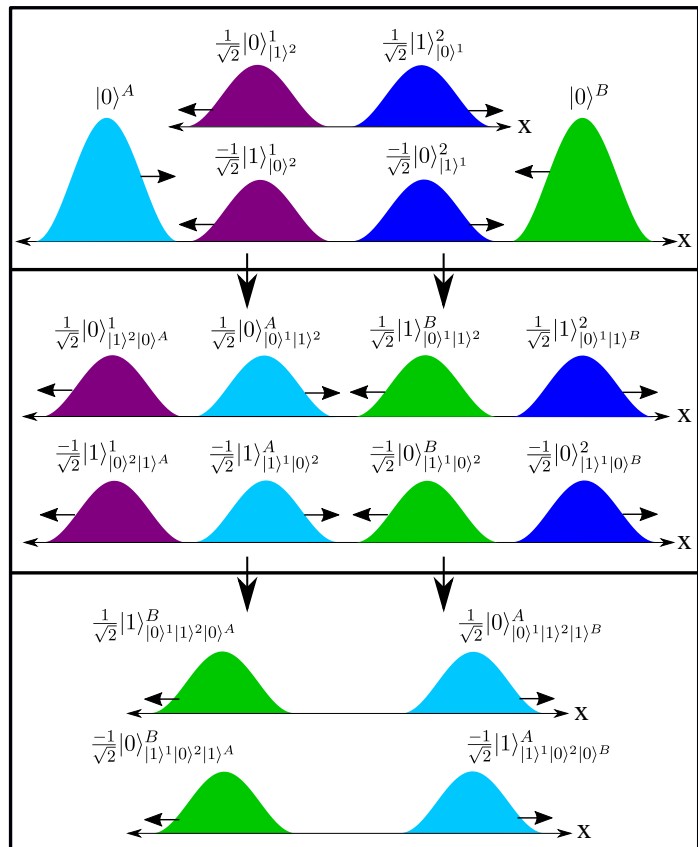

**Figure 6.** Three frames showing the steps of the Mermin–Wigner Bell test, for the case that Alice and Bob measure the same setting. The **top** frame shows the Bell state being sent to Alice and Bob, the **middle** frame is after Alice's and Bob's measurements are completed, and the **bottom** frame is after Alice and Bob meet to share their results. In the experience of each Alice and Bob, the proper entanglement correlations have been obeyed.

### 5.2. Case 2

The situation for Alice's measurement of system 1 is the same as in Case 1. For Bob's measurement, with the same ready state $|0\rangle^B$, a viable unitary is,

$$W^{2B} = |\phi^+\rangle^2|0\rangle^B\langle\phi^+|^2\langle0|^B + |\phi^-\rangle^2|1\rangle^B\langle\phi^-|^2\langle0|^B$$

$$+ |\phi^+\rangle^2|1\rangle^B\langle\phi^+|^2\langle1|^B + |\phi^-\rangle^2|0\rangle^B\langle\phi^-|^2\langle1|^B. \qquad (37)$$

Thus, when Bob measures system 2, the local state carried in the memory of the two systems synchronizes and updates to,

$$W^{2B}U^{12}|0\rangle^1|0\rangle^2|0\rangle^B \qquad (38)$$

$$= \sqrt{\frac{3}{8}}|0\rangle^1|\phi^+\rangle^2|0\rangle^B - \sqrt{\frac{1}{8}}|1\rangle^1|\phi^+\rangle^2|0\rangle^B$$

$$- \sqrt{\frac{1}{8}}|0\rangle^1|\phi^-\rangle^2|1\rangle^B - \sqrt{\frac{3}{8}}|1\rangle^1|\phi^-\rangle^2|1\rangle^B,$$

resulting in the eight wavefunctions,

$$\sqrt{\frac{3}{8}}\psi^B_{0,|0\rangle^1|\phi^+\rangle^2}(x), \qquad -\sqrt{\frac{1}{8}}\psi^B_{0,|1\rangle^1|\phi^+\rangle^2}(x),$$

$$-\sqrt{\frac{1}{8}}\psi^B_{1,|0\rangle^1|\phi^-\rangle^2}(x), \qquad -\sqrt{\frac{3}{8}}\psi^B_{1,|1\rangle^1|\phi^-\rangle^2}(x),$$

$$\sqrt{\frac{3}{8}}\psi^2_{\phi^+,|0\rangle^1|0\rangle^B}(x), \qquad -\sqrt{\frac{1}{8}}\psi^2_{\phi^+,|1\rangle^1|0\rangle^B}(x),$$

$$-\sqrt{\frac{1}{8}}\psi^2_{\phi^-,|0\rangle^1|1\rangle^B}(x), \qquad -\sqrt{\frac{3}{8}}\psi^2_{\phi^-,|1\rangle^1|1\rangle^B}(x).$$

(39)

Now, Alice and Bob meet, and their memories synchronize to the local state,

$$V^{12}W^{2B}U^{12}|0\rangle^1|0\rangle^2|0\rangle^A|0\rangle^B$$

(40)

$$= \sqrt{\frac{3}{8}}|0\rangle^1|\phi^+\rangle^2|0\rangle^A|0\rangle^B - \sqrt{\frac{1}{8}}|1\rangle^1|\phi^+\rangle^2|1\rangle^A|0\rangle^B$$

$$- \sqrt{\frac{1}{8}}|0\rangle^1|\phi^-\rangle^2|0\rangle^A|1\rangle^B - \sqrt{\frac{3}{8}}|1\rangle^1|\phi^-\rangle^2|1\rangle^A|1\rangle^B,$$

resulting in the eight spatial wavefunctions,

$$\sqrt{\frac{3}{8}}\psi^A_{0,|0\rangle^1|\phi^+\rangle^2|0\rangle^B}(x), \qquad -\sqrt{\frac{1}{8}}\psi^A_{1,|1\rangle^1|\phi^+\rangle^2|0\rangle^B}(x),$$

$$-\sqrt{\frac{1}{8}}\psi^A_{0,|0\rangle^1|\phi^-\rangle^2|1\rangle^B}(x), \qquad -\sqrt{\frac{3}{8}}\psi^A_{1,|1\rangle^1|\phi^-\rangle^2|1\rangle^B}(x).$$

$$\sqrt{\frac{3}{8}}\psi^B_{0,|0\rangle^1|\phi^+\rangle^2|0\rangle^A}(x), \qquad -\sqrt{\frac{1}{8}}\psi^B_{0,|1\rangle^1|\phi^+\rangle^2|1\rangle^A}(x),$$

$$-\sqrt{\frac{1}{8}}\psi^B_{1,|0\rangle^1|\phi^-\rangle^2|0\rangle^A}(x), \qquad -\sqrt{\frac{3}{8}}\psi^B_{1,|1\rangle^1|\phi^-\rangle^2|1\rangle^A}(x).$$

(41)

It is again clear from these final wavefunctions that the entanglement correlations for the Bell state have been correctly obeyed for the case that the measurement settings were not aligned. The steps are shown in Figure 7.

The two cases together show that the local fluid model exactly reproduces all of the empirical predictions of standard nonlocal quantum mechanics for this Bell test.

It is important to note that the entanglement correlations are not obeyed in any meaningful sense until Alice and Bob meet, their memories synchronize, and their wavefunctions are paired by their indexes. Prior to that, there were Alices in space-time who had experienced either outcome, and also Bobs who had experienced either outcome, but there is no correlation among them, which is clear because their distributions in space-time always match their respective reduced density matrices.

There is also a general lesson here about post-selected ensembles of quantum measurements. All of the measurement outcomes exist as wavefunctions with different indexes, with an overall distribution still given by the reduced density matrix of that system. When a single observer locally collects data from the post-selected system and the other systems of interest, this is where the entanglement correlations associated with the post-selection are realized. The observers who saw the desired post-selection will also see the anticipated entanglement correlations. Examples include spontaneous parametric down conversion of entangled states [40,41], delayed-choice quantum erasure [42], measurements of weak values [43], and the Delft Bell experiment [44].

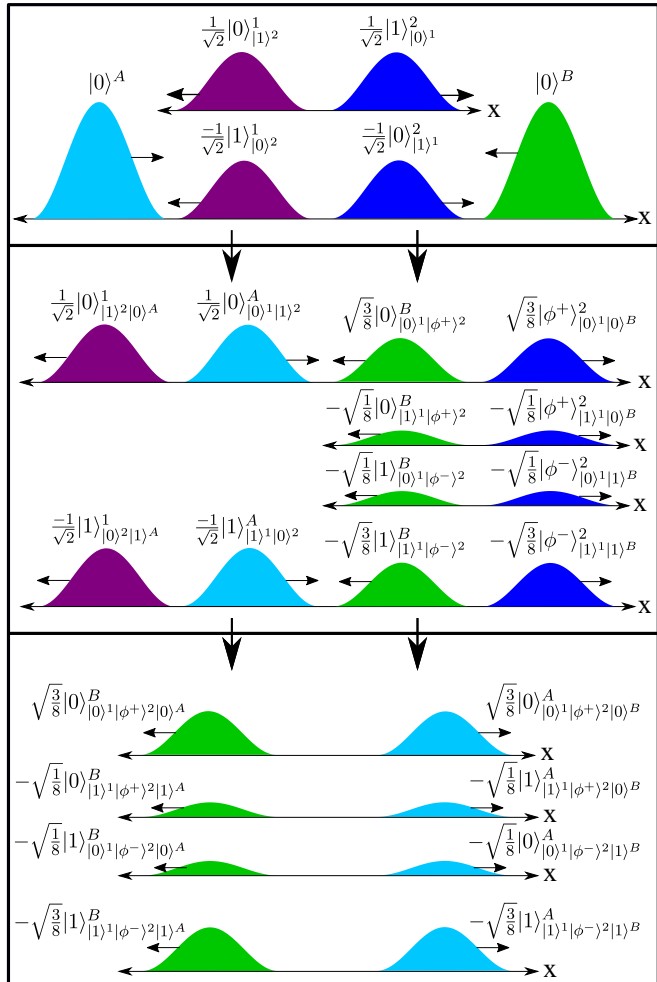

**Figure 7.** Three frames showing the steps of the Mermin-Wigner Bell test, for the case that Alice and Bob measure the different settings. The **top** frame shows the Bell state being sent to Alice and Bob, the **middle** frame is after Alice's and Bob's measurements are completed, and the **bottom** frame is after Alice and Bob meet to share their results. In the experience of each Alice and Bob, the proper entanglement correlations have been obeyed.

*5.3. Demonstrating the Local Hidden Variables*

As mentioned in the introduction, the empirical facts of this theory lead to a picture with many-worlds. It is worth emphasizing again that the treatment we have just given is an explicit local hidden variable model of the Bell experiments, that successfully reproduces the entanglement correlations in a single-space-time (*one-world* is an often-unstated assumption of Bell's theorem, which is violated here).

To make this completely undeniable, we consider a quick demonstration using students which makes it clear that all entanglement correlations and Born rule statistics are obeyed, and everything happens on world-lines in a single space-time, with a Lorentz invariant causal structure.

We will have 8 students in one room who play different copies of Alice, and 8 more in another room who play different copies of Bob, with each group receiving one of the qubits from the singlet state. Students playing the same person are understood to not observe each other. The students in each room collectively choose one of the three settings, and then they all measure that same setting. The results is that a random 4 of the students in that room get 'up' and the other four get 'down', consistent with the 50% Born rule probability for the reduced density matrix of the Bell state available in each room. Each student writes their chosen setting and their result on a sign they then carry, but they are still completely separated in their different rooms, and have not communicated in any way.

The students then gather in a single room with absolutely no light, so they cannot see the other students' signs. A referee with night-vision goggles then pairs the students up (one Alice and one Bob), and then sends these pairs out of the room, where the Alice and Bob meet and look at each other's signs. In both cases, the students will always find that the Born rule probabilities from the singlet state were obeyed, regardless of their settings.

In Case 1, the referee does this by pairing the four 'up' Alices with the four 'down' Bobs, and vice versa, so all eight Alices meet a Bob with the opposite spin, as expected for the singlet state.

In Case 2, the referee pairs one 'up' Alice with an 'up' Bob, one 'down' Alice with a 'down' Bob, three 'up' Alices with three 'down' Bobs, and three 'down' Alices with three 'up' Bobs. The fraction of students with each outcome thus reproduce the Born rule statistics, so in a large ensemble of identical trials, the students will experience them as frequentist probabilities.

To help visualize this, we can think of the original Alice wave packet in Figure 6 as containing 8 students, who divide up into 2 groups of 4 when Alice measures qubit 1. There are likewise 8 students in the original Bob packet, which divide up into 2 groups of 4. When Alice and Bob meet, the 4 from each group are paired off as indicated in the cases above, depending on what settings each group chose. Figure 7 for Case 2 can also be broken up using the 16 students.

## 6. Single-Particle Unitaries and Spatial Superpositions

As discussed above, all single-particle unitary operations on a system really correspond to weak entanglement in a standard two-system interaction. This is made explicit here. The control system may be macroscopic, but we treat it as a single quantum system in an environmentally decohered basis. After the interaction, the entangled state in local memory contains orthogonal terms for the target system and terms that are nearly indistinguishable for the control system. To get the single-particle approximation, the observer (environment) measures the control system in the same decoherent basis it began in, which results in multiple nearly identical wavefunctions, each having undergone approximately the intended single-particle unitary. The single-particle unitary approximation is to ignore the differences between these wavefunctions and treat them as one (dropping the indexes corresponding to the control system).

For two spins, the initial state of the target system is $|0\rangle^t$, and for the control system it is $|0\rangle^c$. After the interaction, the local state carried in each system's memory is $a|0\rangle^c|0\rangle^t + b(\cos\epsilon|0\rangle^c + \sin\epsilon|1\rangle^c)|1\rangle^t)$, for $|\epsilon| \ll 1$.

The six wavefunctions of the two systems are

$$a\psi^t_{0,|0\rangle^c}(x), \qquad a\psi^c_{0,|0\rangle^t}(x),$$

$$b\cos\epsilon\,\psi^t_{1,|0\rangle^1}(x), \quad b\cos\epsilon\,\psi^c_{0,|1\rangle^t}(x), \tag{42}$$

$$b\sin\epsilon\,\psi^t_{1,|1\rangle^1}(x), \quad b\sin\epsilon\,\psi^c_{1,|0\rangle^t}(x),$$

The experimenter begins in state $|0\rangle^e$. The experimenter now measures the control system in the binary basis, resulting in local state

$$a|0\rangle^t|0\rangle^c|0\rangle^e + b|1\rangle^t(\cos\epsilon|0\rangle^c|0\rangle^e + \sin\epsilon|1\rangle^c|1\rangle^e)) \tag{43}$$

$$\approx (a|0\rangle^t + b|1\rangle^t)|0\rangle^c|0\rangle^e = U^t|0\rangle^t|0\rangle^c|0\rangle^e$$

in both systems' memories. The state of the target system has effectively undergone single-system unitary $U^t$, and the states of the control and experimenter systems are unchanged. Under this approximation, only the memory of the target system is updated from $|0\rangle^t$ to $U^t|0\rangle^t = a|0\rangle^t + b|1\rangle^t$, and such single-system unitaries must be included when memories synchronize during local interactions.

This simple treatment for spin systems can be easily generalized to any pair of systems whose interaction results in a weakly entangled state.

### 6.1. The Beam Splitter and Einstein's Objection to Nonlocal Collapse

Although the full treatment of entangled infinite-dimensional systems in space-time is quite complicated, we can still get a good idea what is going on for cases where only a finite number of spatial modes need to be considered.

The simplest example is a particle incident on a beam splitter, where we use the single-system unitary approximation. After the beam splitter the internal memory for the superposed particle state on paths I and II is $Ua_I^\dagger|0\rangle = \frac{1}{\sqrt{2}}(a_I^\dagger + a_{II}^\dagger)|0\rangle = \frac{1}{\sqrt{2}}(|1\rangle^I|0\rangle^{II} + |0\rangle^I|1\rangle^{II})$ in the Fock basis. Treating the vacuum mode as a local state, this corresponds to the four wavefunctions,

$$\frac{1}{\sqrt{2}}\psi(x,t)^I_{1,|0\rangle^{II}}, \quad \frac{1}{\sqrt{2}}\psi(x,t)^I_{0,|1\rangle^{II}},$$

$$\frac{1}{\sqrt{2}}\psi(x,t)^{II}_{1,|0\rangle^I}, \quad \frac{1}{\sqrt{2}}\psi(x,t)^{II}_{0,|1\rangle^I}, \tag{44}$$

where it is implicit that $\psi(x,t)^I$ and $\psi(x,t)^{II}$ are two different wavefunctions, evolving along two different paths. For a given system, the fluid from different paths can mix and interfere locally, consistent with the local evolution of the internal memory state.

As an aside, we can see that for a single-system unitary situation with a continuous path degree of freedom like a single-slit diffraction, the local state in internal memory becomes,

$$\int_{\{x\}} \phi(x)a_x^\dagger|0\rangle dx = \int_{\{x\}} \phi(x)|1\rangle^x \bigotimes_j^{\{x\}-x} |0\rangle^j dx \tag{45}$$

where $\{x\}$ is the set of all paths, and the interaction produces some normalized distribution $\int_{\{x\}} |\phi(x)|^2 dx = 1$ over all of the paths. We then have an infinite number of spatial wavefunctions for each specific path $x_0$,

$$\psi(\mathbf{x},t)^{x_0}_{\left(1, \otimes_j^{\{x\}-x_0}|0\rangle^j\right)}\phi(x_0),$$

$$\psi(\mathbf{x},t)^{x_0}_{\left(0,|1\rangle^{x_1} \otimes_j^{\{x\}-x_0-x_1}|0\rangle^j\right)}\phi(x_1), \tag{46}$$

where $x_1$ is any path other than $x_0$, and where each $\psi(\mathbf{x},t)^{x_0}_i$ is a distinct wavefunction that evolves on path $x_0$, which break down into one case where the particle is on path $x_0$ (Fock state $|1\rangle^{x_0}$, upper Equation (46)), and infinitely many others where there is vacuum on path $x_0$ (Fock State $|0\rangle^{x_0}$, lower Equation (46)), because the particle is on path $x_1$. The fluid on the different paths can mix and interfere if the paths meet locally, just as in the 2-path case. In the spin cases analyzed above, there is only one path, and the vacuum modes have zero amplitude. We won't spend any more time on continuous degrees of freedom here, but this discussion is included to emphasize the generality of the present theory.

Now, returning to the beam splitter, we have a simple tool to demonstrate how the present model resolves Einstein's objection at the 1927 Solvay conference to the instantaneous and nonlocal nature of wavefunction collapse in the emerging quantum theory. We have already explained how the experience of collapse and Born rule probabilities arise for the individual fluid particles along their world-lines, so this is just a matter of applying these principles. The situation is analogous to Case 1 in the Bell test.

Suppose we send a particle through a beam splitter, and then path I leads to Alice's detector, and path II to Bob's space-like separated detector. After the particle is detected, Alice and Bob meet to compare results, and they always find that only one of them has detected the particle. Roughly speaking, Einstein's objection was that in a single objective

world where a spatially superposed wavefunction causally mediates between the source and detectors, once Alice detects the particle, something must instantaneously prevent the wavefunction from also triggering Bob's detector, which violates local causality.

In the present local model, half of the fluid goes to Alice and the other half to Bob. Alice branches into two subgroups of fluid; those that detected the particle and those that did not. Bob likewise branches into two subgroups, with the indexes reversed from Alice. When they meet, the Alices who detected the particle have matched indexes with the Bobs who did not, and vice versa, and so they always find that only one of them has detected the particle, as expected.

Once Alice's detector on path I has either fired or not, the local state in her memory updates to $\frac{1}{\sqrt{2}}\left(|1\rangle^{\mathrm{I}}|0\rangle^{\mathrm{II}}|1\rangle^{A} + |0\rangle^{\mathrm{I}}|1\rangle^{\mathrm{II}}|0\rangle^{A}\right)$, where $|1\rangle^{A}$ indicates her detector has fired, and she has the two spatial wavefunctions,

$$\frac{1}{\sqrt{2}}\psi(x,t)^{A}_{1,|1\rangle^{\mathrm{I}}|0\rangle^{\mathrm{II}}}, \quad \frac{1}{\sqrt{2}}\psi(x,t)^{A}_{0,|0\rangle^{\mathrm{I}}|1\rangle^{\mathrm{II}}}. \tag{47}$$

Likewise for Bob's detector on path II, the local state in memory updates to $\frac{1}{\sqrt{2}}\left(|1\rangle^{\mathrm{I}}|0\rangle^{\mathrm{II}}|0\rangle^{B} + |0\rangle^{\mathrm{I}}|1\rangle^{\mathrm{II}}|1\rangle^{B}\right)$, and his two wavefunctions are,

$$\frac{1}{\sqrt{2}}\psi(x,t)^{B}_{0,|1\rangle^{\mathrm{I}}|0\rangle^{\mathrm{II}}}, \quad \frac{1}{\sqrt{2}}\psi(x,t)^{B}_{1,|0\rangle^{\mathrm{I}}|1\rangle^{\mathrm{II}}}. \tag{48}$$

Finally, when Alice and Bob meet, the local state carried in both their memories synchronizes to

$$\frac{1}{\sqrt{2}}\left(|1\rangle^{\mathrm{I}}|0\rangle^{\mathrm{II}}|1\rangle^{A}|0\rangle^{B} + |0\rangle^{\mathrm{I}}|1\rangle^{\mathrm{II}}|0\rangle^{A}|1\rangle^{B}\right), \tag{49}$$

and we have the four expected wavefunctions, where in each case, either only Alice or only Bob has detected the particle,

$$\frac{1}{\sqrt{2}}\psi(x,t)^{A}_{1,|1\rangle^{\mathrm{I}}|0\rangle^{\mathrm{II}}|0\rangle^{B}}, \quad \frac{1}{\sqrt{2}}\psi(x,t)^{A}_{0,|0\rangle^{\mathrm{I}}|1\rangle^{\mathrm{II}}|1\rangle^{B}}$$

$$\frac{1}{\sqrt{2}}\psi(x,t)^{B}_{0,|1\rangle^{\mathrm{I}}|0\rangle^{\mathrm{II}}|1\rangle^{A}}, \quad \frac{1}{\sqrt{2}}\psi(x,t)^{B}_{1,|0\rangle^{\mathrm{I}}|1\rangle^{\mathrm{II}}|0\rangle^{A}}. \tag{50}$$

### 6.2. Stern-Gerlach Devices

The true function of a Stern-Gerlach device [45] in the local model involves many force-carrying particles being emitted locally from the magnet and then propagating to the spin and interacting locally with it. Here we approximate that entire process by a single local interaction unitary and boundary condition, using the single-system unitary approximation, and treating just two output modes.

In conventional quantum theory, the incoming state will be $\left(a|0\rangle^{s} + b|1\rangle^{s}\right)|1\rangle^{\mathrm{I}}|0\rangle^{\mathrm{II}}$, and the action of the magnetic field will be to transmit the $|0\rangle^{s}$ state and reflect the $|1\rangle^{s}$ state, causing the path and spin to become entangled, and the state to become, $\left(a|0\rangle^{s}|1\rangle^{\mathrm{I}}|0\rangle^{\mathrm{II}} + b|1\rangle^{s}|0\rangle^{\mathrm{I}}|1\rangle^{\mathrm{II}}\right)$. The process is shown in Figure 8

In the present quantum theory, there are initially two identical spatial wavefunctions, $a\psi^{s}_{0}(x,t)$ and $b\psi^{s}_{1}(x,t)$ which move on path I, and after the interaction they have evolved to,

$$a\psi^{s}_{0,|1\rangle^{\mathrm{I}}|0\rangle^{\mathrm{II}}}(x,t), \quad b\psi^{s}_{1,|0\rangle^{\mathrm{I}}|1\rangle^{\mathrm{II}}}(x,t),$$

$$a\psi^{\mathrm{I}}_{1,|0\rangle^{s}|0\rangle^{\mathrm{II}}}(x,t), \quad b\psi^{\mathrm{I}}_{0,|1\rangle^{s}|1\rangle^{\mathrm{II}}}(x,t), \tag{51}$$

$$a\psi^{\mathrm{II}}_{0,|1\rangle^{s}|1\rangle^{\mathrm{I}}}(x,t), \quad b\psi^{\mathrm{II}}_{1,|0\rangle^{s}|0\rangle^{\mathrm{I}}}(x,t)$$

where $\psi^{\mathrm{I}_{s}}_{i}(x,t)$ is a wavefunction that propagates along path I as it evolves, and $\psi^{\mathrm{II}_{s}}_{i}(x,t)$ is a different wavefunction that propagates along path II.

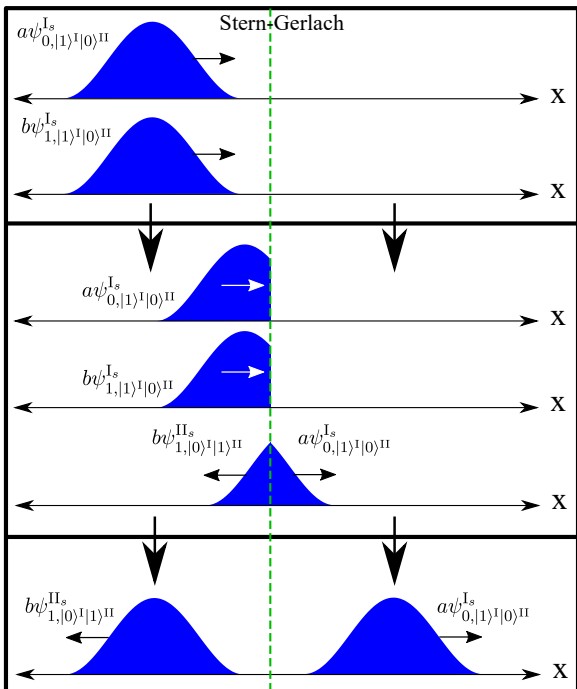

**Figure 8.** Three frames showing the local entanglement of the spin and path degrees of freedom as particle *s* passes through a Stern-Gerlach device, approximated as a point (vertical dotted line), which either transmits or reflects the particle. The incoming wavefunction $\psi^{\mathrm{I}_s}(x,t)$ is identical for both spin states prior to this entanglement (as in all previous examples in this article). After the interaction, there are two different wavefunctions, $a\psi^{\mathrm{I}_s}_{0,|1\rangle^{\mathrm{I}}|0\rangle^{\mathrm{II}}}(x,t)$ continuing in the same direction and $b\psi^{\mathrm{II}_s}_{1,|0\rangle^{\mathrm{I}}|1\rangle^{\mathrm{II}}}(x,t)$ along a different direction. The interference between the incoming and reflected waves is not shown. The vacuum modes are also omitted for clarity.

## 7. Conclusions

The local space-time quantum fluid model presented here fully supplants the configuration-space Schrödinger-picture quantum mechanics for multiple particles, and reproduces all of the standard predictions. The relativistic generalization of the fluid model, following Schwinger's covariant formulation, should only contain world-lines as particle trajectories, which results in a local Lorentz invariant causal structure.

We expect this will require a correction to the coarse-grained single-particle Schrödinger equation even for nonrelativistic energies, to prevent superluminal signalling [46,47]. It may also require a correction of the Dirac equation for the same reason, but this is less clear. Either way, these equations are still quite close to the true (unknown) equations of motion that should underlie this model, and if we use them for all of our single-particle evolution, and the same coupling unitaries for our local boundary conditions, then the new model makes identical predictions.

These details notwithstanding, we now have a quantum theory compatible with the local Heisenberg-Schrödinger picture that Schwinger called the 'interaction representation'. This theory is consistent with the local Heisenberg treatment used in relativistic quantum field theory and the Standard Model, while delocalized Hilbert/configuration space treatments are not.

That said, there are clearly many situations where the configuration space wavefunction is a useful tool for calculations, but it is truly only a delocalized approximation of the proper local physics. This calls into question every development in the foundations of quantum mechanics based on this delocalized treatment of entanglement. The fact that this was not better understood in the 1950s seems baffling at first, but when one considers the historical context, the picture starts to becomes clear.

First, Bohr and Heisenberg's *complementarity* had made the pursuit of any realist interpretation of quantum mechanics with a clear narrative unpopular. Ideas like this started to be denigrated as 'philosophy' rather than 'physics', and students were taught to 'shut up and calculate.' It was no longer encouraged for physicists to know what they were talking about, so long as their mathematics led to accurate predictions.

Second, the very successful formalism of quantum field theory that developed in that environment makes use of both past and future boundary conditions, and the mathematics can be interpreted as retrocausal effects propagating from future to past. Notions of propagation from past to future, or even descriptions of what is happening between distant past and future boundary conditions, are heavily obscured in the mathematical and conceptual machinery of these theories - particularly in the path integral formalism. Furthermore, the plane wave solutions of the Dirac and Klein-Gordon equations are delocalized, filling all of space, and local packets are treated as emerging from their interference. The mathematics of the theory are delocalized in both space and time, and there is simply no clear physical narrative of what is going on.

Third, Bell's theorem has had a much more recent impact on the community, and has created the widespread and mistaken impression that any local realist interpretation of quantum mechanics is impossible. What Bell's theorem actually proves is that a local theory must be either superdeterministic, or have multiple copies of each observer, who may experience different outcomes from the measurement, but who each experience just one—exactly like the multiple perspectives of quantum fluid particles on different world-lines in space-time. QED has always been a local many worlds theory of this type, but this fact was obscured by the lack of a proper interpretation.

All told, it is easy to see why the pursuit of a local realist narrative in space-time has not been a high priority in the foundations community, but this appears to have been a colossal mistake. In particular, it has led to the idea that we must abandon the notion of definite causal order, especially at the interface between quantum mechanics and relativity. Very few people seem to understand that there is already a covariant local realist theory hidden away in the Standard Model.

Finally, it may be possible to extend this model to include a local ballistic treatment of quantum gravity in a single space-time with a fixed shape. This is not to say that we can provide a complete theory at present, but the treatment for gravitons as local force-carriers of gravity should be fundamentally similar to the treatment of photons as local force-carries of electromagnetism, but with an aspect that affects the rates of local clocks. This model, with its many local perspectives in space-time, might then untangle the issues of causal structure, and allow quantum theory and relativity theory to be fully integrated. Even if gravity actually does affect the shape of space-time, we could still have a quantum theory of fluid particles on world-lines in different branched space-times, which should still have a definite causal structure.

**Supplementary Materials:** The following supporting information can be downloaded at: https://www.mdpi.com/article/10.3390/quantum5010011/s1. Supplemental Information: Local quantum theory with fluids in space-time. References [48–54] are cited in the supplementary materials.

**Funding:** This research received no external funding.

**Acknowledgments:** Thanks to Justin Dressel, Kelvin McQueen, Travis Norsen, and Alex Matzkin for many helpful discussions. Special thanks to an anonymous reviewer who helped me to correct errors in two earlier version of this article, which led to this simplified picture.

**Conflicts of Interest:** The authors declare no conflict of interest.

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
