# Peer review of "Local Quantum Theory with Fluids in Space-Time"

_quantumrep, doi:10.3390/quantum5010011_

Round 1

Reviewer 1 Report

Report on “Local quantum theory with fluids in space-time”

The paper wishes to propose a new local, many-worlds. Lorentz-invariant, theory with a spatio-temporal ontology as an alternative to standard quantum theory building on Schwinger’s 1948 proposal.

 In virtue of such a proposal, the paper is very interesting, as such a theory, perhaps aside from being a many-worlds theory, seems to be the Holy Grail of physics, combining quantum nonlocality with relativity in spacetime.  

Nonetheless, the paper is not very clear about what the theory is. (Hence, the following comments will have to do with the clarity of the presentation, and do not deal with the mathematical and physical soundness of the proposal, which should be judged *after* a clearer reformulation.)  Since the proposal aims at completely supplant standard quantum mechanics (and to solve the measurement problem, in addition to make the theory relativistic), the presentation of the theory, I think, should not rely on the standard formalism without clarification. Instead, the paper entirely relies on the quantum formalism, and this is, in my opinion, obscuring its point.  

It should be clear(er) what the ontology is. The paper reads:’ In the present model, all (quantum) systems are comprised by pseudo-classical fluids in a single objective locally-Minkowski space-time and

the classical particles in these fluids follow worldlines through that space-time” (page 2). To me, this means that the theory has a particle ontology (even if I am slightly confused by the label ‘classical’: does it mean that they nature is the same as the particles in classical mechanics? But in quantum mechanics there are no particles, so I do not see the necessity of the precisification), and that these particles behave in a way which is approximately described by the motion of a (nonclassical) fluid.    

If so, then why not say it more clearly? Also, why do not have this statement followed by a clear equation of motion for the particles? It presumably would include something like the state vector of the wave function, or the like. And in this way it would be clear what their meaning is (in my understanding, they are dynamical entities. But am I correct? ). Instead in the paper one talks about these entities without really clarifying where they are coming from.

Similar remarks concerning clarity apply here: “We will not be working with the individual trajectories of the classical particles in the fluids here, since we do not yet know how to choose a unique solution” (pp. 2-3).

The solution of which equation are we talking about? Why we do not have such a solution?  

 Continuing: “Here we consider the continuum fluid equations obtained by coarse-graining over the ballistic trajectories of the individual particles comprising the fluid “

It would be helpful to clarify what the locution “ballistic trajectories” is appropriate here, and why this amounts to a coarse graining.

More generally, in the rest of the paper, I do not appreciate the chosen language in terms of “external and internal memory tapes” and “nature computing” or “having access to” such states, because I find it confusing. While I agree that anthropomorphic language can often be useful, I do not think this is the case. Since we are currently at the stage of formulating *precisely* a fundamental physical theory which aims at substituting standard quantum theory (which was already bad in this respect), we should have equations describing the ontology’s spatiotemporal behavior, instead of these vague terms. What I am trying to say is that if Bell was right that no fundamental physical theory should be formulated in terms of imprecise notions, then this theory seems to be in trouble because this is what seems to be happening here. Let me be clear: I think that the theory *can* be made precise, avoiding the above mentioned locutions to favor a presentation in terms of ontology and its dynamics. So, I do not see a fundamental objection based on that. However, starting with a theory which is vaguely formulate does not seem to be the right way to proceed.  

 A remark/question on the discussion of the preferred basis. The paper reads:” Macroscopic systems are truly composed of many fundamental single-particle systems, each with its own fluid and set of single-particle wavefunctions, but in many cases the correct intuition can be obtained by approximating the macroscopic system as a single fluid, whose particles are different copies of the whole system” (p. 4). I do not understand how macroscopic object might not have a preferred basis. If we have a particle fundamental ontology, the preferred basis is automatically selected as being the position basis.

Author Response

I thank the reviewer for their time and expertise, and for their generally supportive comments regarding this work.  I respond to the reviewer’s comments below, and I have made significant improvements to the introduction of the article in response to many of them.

First off, the reviewer was critical of the fact that the paper relies heavily on the standard quantum formalism, but this is unavoidable because the new formalism incorporates the standard Hilbert space formalism of quantum mechanics, but with a different local Hilbert space state at each event in space-time, built only using causal information in the past light cone of the event, rather than a single one from the present state of the entire universe.  Furthermore, the set of indexed single-particle wavefunctions in space-time are nearly identical to the single-particle wavefunctions in the standard formalism, which are patched together into a wave-field for each fundamental quantum system.  Each system’s wave-field describes a single conserved fluid in space-time, with boundaries where multiple systems interact and the set of local indexes changes.  The set of indexes, and the proportions of fluid with each index value, are described by the local Hilbert space state (the internal memory), so the Hilbert space states and spatial wavefunctions are integrally connected in this theory.  Thus, while I appreciate the reviewer’s suggestion to improve the clarity of the presentation, I cannot really remove these elements of standard quantum theory from the new theory.

The reviewer’s next question concerns the ontology of the theory.  In this theory, the probability density of a standard single-particle wavefunction (the mod square) is re-interpreted as the local density of a literal fluid in space-time, and the conservation of probability current is re-interpreted as a conservation of fluid current.  Thus, a single electron with a spatially distributed wavefunction, is interpreted as an entire fluid, made of a countably infinite number of fluid particles, each of which is like a classical point particle on a world-line.  I have added some of the above to the manuscript in an effort to clarify the ontology.  The standard quantum evolution of the single electron’s wavefunction thus describes the collective evolution of this entire fluid, but not the individual trajectories of the fluid particles.  This is the sense in which the present treatment is coarse-grained – it describes only the local density and current density of the fluid, but not the individual trajectories.  At the present stage of the theory’s development, we do not have a unique solution for the trajectories.  The Bohmian trajectories prove that such solutions are possible (e.g. the velocity field is the ratio of the standard probability current and density), but there are many other possible sets of trajectories which could also be consistent with the same total current and density.  I have updated this part of the introduction in the hope that this comes across more clearly.  I hope that this has explained some of the details that the reviewer found unclear about the fluid ontology.

Now, on the reviewer’s comments about language, I appreciate the advice, and I have changed the language of the paper so that it is not anthropomorphic.  I have dropped the ‘tape’ from the discussion of the memories, although the internal external memories are necessary technical elements of the theory – the internal memory is a local Hilbert space state, and each external memory corresponds to a single empirical outcome for a macroscopic measurement.  I have changed ‘ballistic trajectories’ to simply `trajectories’ to avoid needless confusion.  I have replaced the language about nature performing a computation, by saying that the memories of two particles that meet at an event undergo a joint evolution.  I have also changed the language regarding the inaccessibility of the internal memory to say simply that it cannot be directly measured, and clarified that this means it does not relate to a local energy density, in contrast to the case of measurable types of physical memory, which do.

I hope that these clarifications of language and added explicit examples will resolve the issues the reviewer found to be vague.  The aspects of the theory that are uniquely pinned down are all precisely specified, and I still hope to pin down the specific trajectories in future work.

The reviewer’s final comment on macroscopic systems was helpful, since I realized I had not made it clear enough that macroscopic systems generally do have a preferred basis, which is the basis containing the measurement outcomes they have observed.  The issue of the preferred basis for microscopic systems is a bit more subtle.  Even though the fluid particles each have definite positions and velocities, the indexed packets can be in any basis.  To make this clear, consider an electron prepared in a superposition of the ground state and first excited state of a harmonic oscillator.  If an observer measures the energy of the electron, this will result in an entangled state with two orthogonal terms in the macroscopic preferred (energy) basis, and the fluid particles of both the electron and observer will also be divided into two separate indexed packets corresponding to the two possible outcome states in that basis.  Since there is only a preferred basis for macroscopic systems, position is only a preferred basis for an electron when it has been measured by a macroscopic observer.  In this model, one must be careful not to conflate the behavior of entire wavefunctions (packets) with the behavior of individual fluid particles.

Reviewer 2 Report

The author investigated the local space-time quantum fluid model presented fully supplants configuration-space Schrodinger-picture quantum mechanics for multiple particles and reproduces all the standard predictions. 

The paper needs to add more mathematical detail, which it is very short.

The language minor spell check.

The references are written incorrectly.

Author Response

I thank the reviewer for their time and expertise.  Their comments are relatively brief, but I have tried to improve the article in some of the ways suggested.  I respond to the comments below.

With regard to mathematical detail, I have added some additional math describing how the probability density and probability current are re-interpreted as literal fluid density and current in this model, where a single quantum system like an electron is actually comprised or a large number of classical-fluid particles which move on world-lines (and the conservation of probability current carries over to conservation of fluid current in this model).  I could not see how to add more detail to the mathematical details of the Hilbert space states which comprise the local memory at each event – this was already specified as a Hilbert space state constructed using initial states and unitaries strictly from within the past light cone of the event, and it has also been demonstrated with many examples how this state gives rise to the set of local indexes on the many single-particle wavefunctions in space-time.  The evolution of these indexed wavefunctions is the standard quantum evolution when they do not interact with other systems, and I have shown how local interactions with other systems take the form of boundary conditions in space-time that connect the differently-indexed single-particle wavefunctions on either side.  If the reviewer would like to see other mathematical details spelled out more carefully, then I would appreciate it they provide me with a list of details I should address.

I have read through the paper several times to look for spelling errors, and I have fixed everything I have found.  However, as a single author, it is easy to read over one’s own mistakes without noticing them.  I am not asking the reviewer to be my editor, but I would appreciate it if they would point out any particularly egregious spelling or grammar errors they noticed.

Lastly, the references are compiled using bibtex, and the style will likely be changed by the publishing journal.  I have found a few small typos involving capital letters, but I could not see any other problems with the references.  Again, if there is some specific issue other than the bibliography style, I would appreciate it if the reviewer could point them out for me.

Round 2

Reviewer 1 Report

I have no further comments for the author.

Reviewer 2 Report

The paper can be accepted for publishing, however, there are some references that need to revise.